# The Proteasome Activators Blm10/PA200 Enhance the Proteasomal Degradation of N-Terminal Huntingtin

**DOI:** 10.3390/biom10111581

**Published:** 2020-11-20

**Authors:** Azzam Aladdin, Yanhua Yao, Ciyu Yang, Günther Kahlert, Marvi Ghani, Nikolett Király, Anita Boratkó, Karen Uray, Gunnar Dittmar, Krisztina Tar

**Affiliations:** 1Department of Medical Chemistry, Faculty of Medicine, University of Debrecen, 4032 Debrecen, Hungary; aladdin@med.unideb.hu (A.A.); marvighani93@gmail.com (M.G.); kiraly.nikolett@med.unideb.hu (N.K.); boratko@med.unideb.hu (A.B.); karen.uray@med.unideb.hu (K.U.); 2Doctoral School of Molecular Medicine, University of Debrecen, 4032 Debrecen, Hungary; 3Department of Biochemistry and Molecular Cell Biology, School of Medicine, Shanghai Jiao Tong University, Shanghai 200025, China; 4Department of Biochemistry, Albert Einstein College of Medicine, Bronx, NY 10460, USA; yuerci@yahoo.com; 5Departments of Pathology, Memorial Sloan Kettering Cancer Center, New York, NY 10065, USA; 6Institute for Systems Biology, Seattle, WA 98109, USA; gkahlert@systemsbiology.org; 7Proteomics of Cellular Signalling, Luxembourg Institute of Health, 1445 Strassen, Luxembourg; 8Department of Life Science and Medicine, University of Luxembourg, 4365 Esch-sur-Alzette, Luxembourg

**Keywords:** Huntington’s disease, huntingtin protein (Htt), Blm10/PA200-proteasomes, degradation, soluble N-Htt, mutant N-Htt aggregates

## Abstract

The Blm10/PA200 family of proteasome activators modulates the peptidase activity of the core particle (20S CP). They participate in opening the 20S CP gate, thus facilitating the degradation of unstructured proteins such as tau and Dnm1 in a ubiquitin- and ATP-independent manner. Furthermore, PA200 also participates in the degradation of acetylated histones. In our study, we use a combination of yeast and human cell systems to investigate the role of Blm10/PA200 in the degradation of N-terminal Huntingtin fragments (N-Htt). We demonstrate that the human PA200 binds to N-Htt. The loss of Blm10 in yeast or PA200 in human cells results in increased mutant N-Htt aggregate formation and elevated cellular toxicity. Furthermore, Blm10 in vitro accelerates the proteasomal degradation of soluble N-Htt. Collectively, our data suggest N-Htt as a new substrate for Blm10/PA200-proteasomes and point to new approaches in Huntington’s disease (HD) research.

## 1. Introduction

The fundamental cause of many neurodegenerative diseases is the formation of misfolded oligomers and aggregates, which eventually leads to cytotoxicity and cellular death [1]. Most of the time, misfolded proteins and aggregates are successfully removed from cells by the two major proteolytic pathways, the ubiquitin-proteasome machinery (UPS) and autophagy [1,2,3,4]. However, in certain neurodegenerative diseases, such as Parkinson’s or Huntington’s disease, the genetic mutations, which are present in specific proteins (α-synuclein and huntingtin protein, respectively) can lead to the formation of larger oligomers and aggregates [5,6,7]. These aberrant proteins are quite resistant to degradation pathways.

Huntington’s disease is caused by a CAG repeat expansion mutation in exon 1 of the huntingtin gene (*HTT*). Thus, the mutant huntingtin protein (mHtt) contains a polyglutamine tract (polyQ) in the N-terminal region [8]. The severity and the age of onset of the disease greatly depend on the number of polyQ repeats [9]. Mutant Htt is ubiquitously expressed but mainly targets the central nervous system, damaging and killing medium spiny neurons [10,11]. Furthermore, although mHtt does not form insoluble oligomers and aggregates in peripheral tissues, the mutant protein perturbs the homeostasis of peripheral tissues, including elevated reactive oxygen species (ROS) production and mitochondrial dysfunction [12]. The clinical manifestation of Huntington’s disease is characterized by motor dysfunction, severe psychiatric problems, including aggression, depression, and suicidal tendencies, and a decline in cognitive functions. Quality control of the cellular proteome is maintained even in HD. However, with aging and disease progression, the cellular proteostasis network becomes impaired and dysfunctional leading to cellular death [6].

The regulation of the proteasome (26S) is defined by the topology of its proteolytic core complex, the core particle (20S CP proteasome) [13,14,15,16]. Degradation of substrate proteins takes place inside the proteolytic chamber of the core particle. The 20S CP is composed of seven α-subunits and seven β-subunits arranged to form four heptameric rings. Three of the seven β-subunits, the β_1_-, β_2_-, and the β_5_-subunits, have proteolytic activities. The proteolytic activity of the proteasome includes caspase-like, trypsin-like, and chymotrypsin-like activities. The UPS can degrade mHtt protein and has been implicated in the pathogenesis of HD [17,18,19,20,21]. Both in vitro and in vivo studies demonstrate that the proteasome inefficiently degrades the long polyglutamine containing proteins, including the N-terminal fragments of mHtt [22], leading to detrimental cellular effects. Hipp et al. suggested that the activity of the proteasome is indirectly inhibited by aggregated mHtt fragments due to their exceeded solubility threshold [23]. According to their results, N-terminal fragments of mutant Htt, composed of the first 17 amino acids of the huntingtin protein and the adjacent polyQ stretch, are neither competitive nor noncompetitive inhibitors of 26S and they do not choke the proteasome. They propose that the accumulation of ubiquitin conjugates in cells is the consequence of limited 26S capacity, which originates from the disrupted protein folding homeostasis caused by mHtt aggregates.

To promote substrate degradation, the core particle of the proteasome interacts with proteasome activators at either one or both ends of the 20S CP to activate the proteasome, which in turn opens the gate and allows specific substrate entry into the core. In addition to the well-characterized 19S regulatory particle (RP/19S/PA700), which is required for ATP- and ubiquitin-dependent substrate unfolding and degradation, alternative activators of the proteasomes have been described. Activators of the PA28 protein family are present in higher eukaryotes [24,25], while the conserved Blm10/PA200 family is present in yeast and mammals, respectively [26,27,28]. Both Blm10 and PA200 mediate ATP- and ubiquitin-independent specific substrate degradation such as unstructured proteins and acetylated histones [29,30,31,32].

Recently, cryo-EM studies revealed that PA200 binding induces gate opening of the 20S CP [33,34]. PA200 binding initiates rearrangements of the central pore regions of the α ring of the 20S CP into an open channel conformation and helps small substrate peptides pass through rather than globular proteins. PA200 binds to active 20S CP and forms both singly capped and doubly capped variants in vitro.

In the present study, we found that PA200 colocalizes and binds to wild-type (wt) and mutant N-Htt. Furthermore, our data establish that the Blm10/PA200 family enhances the proteasomal degradation of nonaggregated, soluble N-Htt in vitro compared to 20S CP. We also demonstrate that loss of Blm10/PA200 leads to bigger aggregates of mutant Htt in size and number, and causes elevated cellular toxicity. In summary, our results add a new substrate to the palette of Blm10/PA200-proteasomes substrates. Enhancing the activity of the proteasome may help cells to overcome the toxicity of expanded Htt polyQ, and might represent a new therapeutic approach to fight HD.

## 2. Materials and Methods

Reagents and materials were purchased from Sigma-Aldrich (St. Louis, MO, USA) unless otherwise stated.

### 2.1. Cell Lines and Cell Culture

The human neuroblastoma SH-SY5Y cell line (European Tissue Culture) was cultured in Dulbecco’s Modified Eagle’s Medium-high glucose (DMEM), supplemented with 10% heat-inactivated fetal bovine serum (FBS) (Gibco/ThermoFisher, Waltham, MA, USA), 2 mM l-glutamine, 1 mM sodium pyruvate solution, and 100 IU/mL penicillin-100 µg/mL streptomycin (Pen-Strep), incubated at 37 °C and 5% CO_2_. The stably depleted cell line for PA200 (shPA200) and the corresponding control with empty vector (pGIPZ-GFP) were maintained under 1.25 µg/mL puromycin selection as previously described [35].

### 2.2. Yeast Strains, Media, and Growth Conditions

Yeast strains were obtained using standard genetic techniques. All strains are isogenic to BY4741 or BY4742 [36] and are S288C derived. Complete gene deletion, promoter exchange, or tag integration were constructed at the genomic locus by homologous recombination using standard techniques [37,38]. Primer sequences are available upon request. Unless otherwise noted, strains were grown at 30 °C in yeast peptone dextrose (YPD) and were harvested at OD_660nm_ ≈ 1 for log phase cells, at OD_660nm_ ≈ 12 for postdiauxic shift (PDS) phase cells, and after 5 days for stat phase cells. Yeast strains used in this study are provided in Table 1.

### 2.3. Yeast Phenotype Analysis

To perform a phenotype analysis of cells expressing N-Htt fragments with either 25 (Htt25Q) or 103 glutamines (Htt103Q), cells were grown overnight in synthetic complete media containing 2% (*v*/*v*) raffinose and lacking the amino acid for plasmid selection. Induction was initiated by the addition of 2% (*v*/*v*) galactose. After 18 h of induction, cells were diluted to OD_660nm_ = 0.4 and spotted onto solid synthetic drop-out media containing either 2% (*v*/*v*) glucose or 2% (*v*/*v*) galactose. Colony growth on the plates was monitored daily.

### 2.4. Plasmids Constructs

The pHM6-Q23 and pHM6-Q74 plasmids were gifts from David Rubinsztein (Addgene plasmid # 40263; http://n2t.net/addgene: 40263; RRID: Addgene_40263, and Addgene plasmid # 40264; http://n2t.net/addgene: 40264; RRID: Addgene_40264) (Addgene, Watertown, MA, USA) [41]. The pGEX-5 plasmids expressing the GST–N-huntingtin protein fusion constructs containing either 18 polyQ (control) or 51 polyQ (toxic) were obtained from Dr. Ron Kopito, Stanford University, CA. These constructs were modified by the addition of a TEV protease cleavage site between the GST and huntingtin protein exon 1 coding regions. The presence of the GST tag maintains both the control and the toxic variants in a soluble form. Upon cleavage of the GST moiety with TEV protease, the protein with 51 polyQ stretches rapidly forms high molecular weight aggregates [23,42]. Control pGEX-4T-2 was a gift from Dr. Anita Boratkó. The pYES2-25Q-CFP or pYES2-103Q-CFP, expressing huntingtin exon1 fragments with 25 or 103 glutamines (25Q or mutant 103Q) fused to GFP under the control of the *GAL1* promoter were kindly provided by Michael Sherman [43].

### 2.5. Immunofluorescence and Image Analysis

For immunofluorescence staining, neuroblastoma cells were cultured on glass coverslips coated with 1% gelatin. The next day, cells were transiently transfected with the pHM6-Q23 to express the wt huntingtin fragments or pHM6-Q74 to express the mutant huntingtin fragments using Lipofectamine 3000 (Thermo Fisher Scientific, Waltham, MA, USA) according to the manufacturer’s protocol. After 48 h transfection, cells were rinsed with 1 X PBS, fixed in 4% paraformaldehyde for 15 min at room temperature (RT), permeabilized with 0.1% Triton X-100 in PBS for 30 min, and blocked with blocking buffer (3% BSA, PBS, and 0.01% Triton X-100) for 1 h. Cells were incubated for 1.5 h at RT with primary antibodies (listed in Table 2) in blocking buffer. Cells were then rinsed three times with PBS with 0.01% Triton X-100 and then incubated for 1.5 h at RT with secondary antibodies (listed in Table 2) in 1% BSA, PBS, and 0.01% Triton X-100. The nuclei were counterstained with 1 µg/mL DAPI. Coverslips were mounted on slides with mounting media (Dabco 33-LV: Mowiol 4-88, 1:50). Images were taken with an SP8 confocal laser scanning microscope (Leica Biosystems, Wetzlar, Germany) using a 63X HC PL Apo oil CS2 objective. Post-acquisition images were adjusted using LAS X v 3.7.1 software (Leica Biosystems, Wetzlar, Germany). Confocal images were analyzed for aggregate number and size by ImageJ software (imagej.nih.gov). A total of 1000 cells with aggregates from each cell line were counted and analyzed.

Pearson’s and Manders’ coefficients were evaluated using the JACoP plugin in ImageJ [44]. The r values are Pearson’s cross-correlation coefficients indicating colocalization of N-Htt and PA200. Manders (M1, M2) are overlap coefficients. M1 determines the fraction of N-Htt overlapping PA200 and M2 determines the fraction of PA200 overlapping N-Htt. We measured M1 and M2 above threshold values to remove the noise and background in the images and optimize M1 and M2 measurements.

Visualization of N-Htt fragments in live yeast cells was achieved by monitoring GFP fluorescence in cells carrying the plasmids pYES25Q-GFP or pYES103Q-GFP. Live-yeast cell fluorescence was monitored using a fluorescence microscope Olympus BX61 (Olympus Corporation, Tokyo, Japan) at the Albert Einstein Imaging Facility with a 60X or 100X NA 1.4 objective (PlanApo). Fluorescence or differential interference contrast (DIC) images were captured with a cooled CCD camera (Sensicam QE, Cooke, MI, USA) using IPlab 4.0 software (BD Biosciences, San Jose, CA, USA). Images were identically processed using ImageJ software 1.42q. For excitation of Htt25Q-GFP and Htt103Q-GFP, a 425–445 nm band pass filter was used. Emitted light was detected with a 476 nm long pass filter (filter set Olympus) (Olympus Corporation, Tokyo, Japan).

### 2.6. Bacterial Expression and GST Pull-Down Assay

*Escherichia coli* BL21 (DE3) cells transformed with pGEX-4T-2, huntingtin Q18 encoding pGEX-5, or huntingtin Q51 encoding pGEX-5 constructs were induced with 1 mM IPTG and grown at room temperature (RT) with shaking for 3 h at 180 rpm. Cells were harvested by centrifugation and sonicated in lysis buffer (50 mM Tris–HCl (pH 7.5), 0.1% Tween 20, 0.2% 2-mercaptoethanol, protease inhibitors). GST-tagged proteins were isolated by affinity chromatography on glutathione Sepharose 4B (GE Healthcare Life Sciences, Chicago, IL, USA) according to the manufacturer’s protocol. SH-SY5Y grown in 100-mm plates were washed with 1X ice-cold PBS, scraped, and lysed in 600 μL lysis buffer. The lysates were incubated with GST or GST-fused huntingtin Q18 and Q51 proteins coupled to glutathione Sepharose beads for 16 h at 4 °C using gentle rotation. The beads were washed three times with 1X TBST then GST fused proteins were eluted by boiling the samples in 2X SDS sample buffer and tested for interacting proteins by Western blot.

### 2.7. Filter Trap Assay from Human Neuroblastoma Cells

The filter trap assay was performed to analyze SDS-insoluble aggregates according to the published literature [45,46]. After 48 h of transient transfection with pHM6-HA-Q23 and pHM6-HA-Q74, cells were rinsed with 1X PBS and lysed with lysis buffer (40 mM HEPES, pH 7.5, 50 mM KCl, 1% (*v*/*v*) Triton X-100, 2 mM DTT, 5 mM EDTA) supplemented with 1X cOmplete™, Mini, EDTA-free protease inhibitor cocktail. Cell lysates were fractionated into supernatant (S) and pellet (P) fractions by centrifugation (20 min, 16,000 rpm at 4 °C). Pellets with aggregates were resuspended in 75 µL DNA digestion buffer (40 mM Tris-HCl, pH 8.0, 6 mM MgCl_2_, 10 mM NaCl, 10 mM CaCl_2_), and incubated for 1 h at 37 °C with 30 U DNase I (Zymo Research, Irvine, CA, USA) and 100 µg/mL RNase. Reactions were interrupted by 2X termination buffer (4% SDS, 40 mM EDTA, 100 mM DTT). Samples (10 µg, S and P) were adjusted up to 200 µL by adding filter trap buffer (50 mM Tris-Cl, pH 7.5, 150 mM NaCl (TBS) with 2% SDS). Each sample was filtered through a 0.2 µm nitrocellulose membrane (Bio-Rad Laboratories, Hercules, CA, USA) which was pre-equilibrated in filter trap buffer using a Minifold I 96-well dot-blot system (Whatman GmbH, Dassel, Germany). The dot-blots were washed twice with filter trap wash buffer (TBS with 0.1% SDS). The membrane was removed and blocked for 1 h at RT with Odyssey^®^ blocking buffer (LI-COR, Lincoln, NE, USA), and probed with primary antibody in Odyssey^®^ blocking buffer with 0.1% Tween overnight at 4 °C. Membranes were washed three times for 5 min in TBS with 0.05% Tween then probed with Infrared-labeled (IRDye^®^ 800 CW, IRDye^®^ 680 RD) secondary antibodies in Odyssey^®^ blocking buffer with 0.1% Tween for 1.5 h at RT. After secondary labeling, membranes were washed three times for 5 min in TBS with 0.05% Tween. The dot-blots were scanned using LI-COR ODYSSEY^®^ CLx infrared imaging system (LI-COR, Lincoln, NE, USA). The images and infrared signals were analyzed using Image Studio software v 5.2 (LI-COR, Lincoln, NE, USA).

### 2.8. Filter Retardation Assay and Gradient Gel Analysis from Yeast Cells

The preparation of samples for protein aggregation studies was performed as described previously by Ocampo and Barrientos [47]. Briefly, wt or *blm10∆* cells carrying pYES25Q-GFP or pYES103Q-GFP were grown overnight in synthetic complete media containing 2% (*v*/*v*) raffinose and lacking the amino acid for plasmid selection. For induction, galactose was added at a final concentration of 2% (*v*/*v*) to induce protein expression for 10, 14, and 18 h. The cells were pelleted by centrifugation at 1500× *g* for 10 min and then washed with 1.2 M sorbitol. Cells were resuspended in a cell wall digestion buffer (1.2 M sorbitol, 20 mM K_3_PO_4_, pH 7.4, and 0.6 mg/mL zymolase-100T) and incubated for 30–60 min at 30 °C with gentle shaking. After ≈80% of the cells were converted to spheroplasts, they were diluted by the addition of 1.2 M sorbitol with 20 mM KPO_4_ (pH 7.5) and pelleted by centrifugation at 5500× *g* for 10 min followed by washing two additional times (1.2 M sorbitol, 20 mM KPO_4,_ pH 7.5). The spheroplasts were resuspended in lysis buffer (40 mM Hepes, pH 7.5, 50 mM KCl, 1% (*v*/*v*) Triton X-100, 2 mM DTT, 5 mM EDTA, and 1 mM PMSF) and incubated on ice for 1 h. The upper layer (T, total) was carefully transferred to new tubes and spun down for 10 min at 2000× *g*. After centrifugation, the supernatant (S, supernatant) and the pellet (P, pellet) fractions were separated. The pellet fraction was washed with lysis buffer and resuspended in an appropriate volume of water. Protein concentration was determined for each fraction (T, S, P) with a Bradford protein assay (Bio-Rad Laboratories, Hercules, CA, USA). To quantify the amount of SDS-insoluble protein aggregates a filter retardation assay was performed following the method described by Alberti et al. (2010) with minor modifications [48]. The protein (2, 5, 7, and 10 µg) was solubilized in 200 µL of TBS containing with 0.1% SDS and filtered through a 0.2 µm pore size of cellulose acetate membrane (Whatman GmbH, Dassel, Germany) using a Whatman Vacuum Minifold I System apparatus (Whatman GmbH, Dassel, Germany). Each well was washed twice with TBS. The membrane was blocked with 1X TBS + 0.05% (*v*/*v*) Tween 20 containing 5% (*v*/*v*) dry milk and the retained SDS insoluble aggregates were detected with a polyclonal anti-GFP. The detection of signals was performed with ImageQuant LAS4000 mini system (GE Healthcare Life Sciences, Chicago, IL, USA).

For gradient gel analysis, samples were loaded onto a NuPAGE 3–8% Tris-Acetate gel according to the manufacturer’s protocol (Thermo Fisher Scientific, Waltham, MA, USA). The stacking and separating gels were kept and proteins were transferred to a cellulose acetate membrane (Whatman GmbH, Dassel, Germany). The membrane was blocked with 1X TBS + 0.05% (*v*/*v*) Tween 20 containing 5% (*v*/*v*) dry milk and the retained SDS insoluble aggregates were detected by a polyclonal anti-GFP antibody. The detection of signals was performed with ImageQuant LAS4000 mini system (GE Healthcare Life Sciences, Chicago, IL, USA).

### 2.9. RNA Extraction and cDNA Reverse Transcription

Total RNA was extracted using TRI Reagent (Molecular Research Center, Cincinnati, OH, USA) following the manufacturer’s protocol. Samples were treated with DNase I for 15 min at RT in DNA digestion buffer before the reverse transcription (Zymo Research, Irvine, CA, USA). To perform cDNA synthesis, a High-Capacity cDNA Reverse Transcription Kit (Applied Biosystems, Waltham, MA, USA) was used to reverse transcribe 1 µg total RNA with random primers.

### 2.10. Quantitative Real-Time PCR

Real-time PCR was performed with a Lightcycler 480 thermocycler (Roche Molecular Systems, Basel, Switzerland) using SYBR Green Lo-ROX qPCR master mix (Xceed, IAB, Strašnice, Czech Republic). Cycling conditions were as follows: Stage 1: Preincubation 95 °C 10 min, 1 cycle; Stage 2: Amplification 95 °C 15 s, 60 °C 30 s, for 40 cycles; Stage 3: Melt curve analysis 95 °C 0.05 s, 65 °C 1 min, 97 °C 0 s, 1 cycle; Stage 4: Cooling 40 °C 30 s 1 cycle. Threshold values (C_t_-values) for all replicates were normalized to GAPDH. Each of the biological replicates contained three technical replicates for each gene in each sample. To compare the effect of PA200 depletion, 2^−ΔΔCt^ values were calculated to obtain fold expression levels. The primer list is provided in Table 3.

### 2.11. SDS-PAGE and Western Blot

Cells were washed on ice with PBS and lysed in RIPA buffer (50 mM Tris-HCl pH 7.4, 150 mM NaCl, 0.5% Na-deoxycholate, 2 mM EDTA, 1% NP-40, and 50 mM NaF) with protease inhibitors containing 1 mM benzamidine, 1 mM PMSF, and cOmplete Mini-EDTA-free protease inhibitors (Merck KGaA, Darmstadt, Germany). Cells were homogenized by sonication (three times, 1 s with 15 s breaks on ice) and cell lysates were cleared by centrifugation (10 min, 16,000 rpm at 4 °C). Protein (30 µg) was mixed with Laemmli buffer (60 mM Tris-Cl pH 6.8, 2% SDS, 10% glycerol, 0.01% bromophenol blue, and freshly added 140 mM DTT) and boiled (5 min at 95 °C). Each sample was separated by SDS-PAGE, transferred onto 0.45 µm nitrocellulose membranes (GE Healthcare Life Sciences, Chicago, IL, USA). Blots were immunodetected with antibodies listed in Table 2. Bands signals were enhanced using a chemiluminescent substrate (Santa Cruz Biotechnology, Dallas, TX, USA) and captured by ChemiDoc Imager (Bio-Rad Laboratories, Hercules, CA, USA). Densitometry was performed using Image Lab.v.6.0.1 software (Bio-Rad Laboratories, Hercules, CA, USA).

### 2.12. Purification of Proteasome Complexes

Proteasome core particle (CP), Blm10-CP, and 26S complexes were purified as described previously [27,29]. Cells from yMS476 for the CP purification were grown in 2% glucose medium until O.D. 4–5. Cells from yMS122 for the Blm10-CP and 26S purification were grown first in 2% glucose medium until O.D. 2 and switched to 2% galactose medium for 8 h induction until O.D. 4–5. The cell powder was thawed in one pellet volume of 50 mM Tris, pH 8.0, 50 mM NaCl, 5 mM MgCl, and 1 mM DTT. The same buffer, but supplemented with 10% glycerol, 5 mM ATP, and 1X ATP-regenerating mix (ARS) was used for 26S purification. The thawed cell lysate was centrifuged in a GSA rotor (Sorvall, Newtown, CT) for 30 min at 13,000 rpm, and the pellet was discarded. The cleared lysate was batch-incubated with IgG affinity gel (Cappel) for 1.5 h at 4 °C, and the beads were collected and washed with a solution of 50 mM Tris, pH 7.5, 100 mM NaCl, 5 mM MgCl, and 0.5 mM DTT. Proteasome complexes were eluted after cleavage with tobacco etch viral protease (Invitrogen) and concentrated in a Vivaspin 6 Centrifugal Concentrator of 100 kDa proteins (GE Healthcare). Then, the concentrated proteasome complexes were injected into a Superose 6 10/300 GL column. The complexes were first resolved on 3.5% acrylamide native gels, followed by SDS-PAGE and Coomassie blue staining to assess complex composition and purity of the sample. The same protocol was used to produce doubly and singly capped Blm10-CP. The doubly capped complex comes out earlier when a size-exclusion column was used during purification.

### 2.13. Proteasome (20S CP) Activity Assay

Proteasome (20S CP) activity was measured based on the hydrolysis of the fluorogenic substrate Suc-LLVY-AMC (Enzo) for chymotrypsin-like peptidase activity and was performed according to published protocols [29,49].

### 2.14. In Vitro Protein Degradation Assay

The in vitro degradation assay was performed according to Tar et al. [31]. GST-HttQ18 or GST-Htt51 turnover was tested at 30 °C in a reaction mixture containing 50 mM Tris pH 7.5, 25 mM NaCl, 2.5 mM MgCl_2_, 0.5 mM EDTA, 1 mM DTT, 40 nM HttQ51 or HttQ18, and 12 nM Blm10-CP or 12 nM CP or 12 nM 26S in a final volume of 15 µL for each time point. The reactions were incubated at 30 °C and reactions were terminated at the indicated time points by the addition of 4X sample buffer followed by boiling for 3 min and by immunoblot analysis with an anti-GST antibody (Upstate) for N-Htt Q18 or N-Htt Q51 degradation. Proteasome levels in the reactions were monitored with an anti 20S antibody (BioMol). The protein signals were visualized by ImageQuant LAS 4000 and the intensities of protein bands were quantified using the ImageQuant software (GE Healthcare Life Sciences, Chicago, IL, USA). GST-N-HttQ18 and GST-N-HttQ51 were purified as described previously [50].

### 2.15. Peptidomics Analysis

Peptides were cleaned up using a stage-tip micro column [51] and resuspended in water with 0.1% formic acid (Merck, Darmstadt, Germany). The samples were measured on a Q-Exactive mass spectrometer (Thermo-Fisher, Waltham, MA, USA) coupled to a Proxeon nano-LC system (Thermo-Fisher, Waltham, MA, USA) in data-dependent acquisition mode, selecting the top 10 peaks for HCD fragmentation. A 3-h gradient (solvent A: 5% acetonitrile, 0.1% formic acid; solvent B: 80% acetonitrile, 0.1% formic acid) was applied for the samples using an in-house prepared nano-LC column (0.075 mm × 250 mm, 3 μm Reprosil C_18_, Dr. Maisch GmbH, Germany). A volume of 5 μL sample was injected and the peptides eluted with 3 h gradients of 4–76% ACN and 0.1% formic acid in water at flow rates of 0.25 μL/min. MS acquisition was performed at a resolution of 70,000 in the scan range from 300 to 1700 m/z. Dynamic exclusion was set to 30 s and the normalized collision energy to 26 eV. The mass window for precursor ion selection was set to 2.0 m/z. Data were analyzed using the MaxQuant software. The internal Andromeda search engine was used to search MS2 spectra against the *Saccharomyces cerevisiae* proteome database and the Htt-18Q and Htt51Q sequences of the recombinant Htt proteins containing forward and reverse sequences. The search included the variable modification of methionine oxidation and fixed modification of carbamidomethyl cysteine. The minimal peptide length was set to seven amino acids, with an unspecific digest pattern. The FDR was set to 0.01 for peptide and protein identifications. The data was further analyzed using the R software (www.r-project.org) and the tidyverse analysis package [52].

### 2.16. Statistical Analysis

All quantified data are presented as mean ± standard deviation (SD) of n ≥ 3 experiments. Statistical analysis was performed using unpaired Student’s *t*-test to compare the variance between two data sets only and ANOVA followed by Tukey multiple comparisons test when comparing the variances between more than two data sets. All statistical measurements were performed using GraphPad Prism 8.2.1 software. *p* < 0.05 was considered significant (* indicates *p* < 0.05, ** indicates *p* < 0.01, *** indicates *p* < 0.001, and **** indicates *p* < 0.0001).

## 3. Results

### 3.1. Blm10-Proteasomes Antagonize the Toxicity of Mutant Huntingtin Expression in Yeast

HD is an age-related disorder where cells expressing polyQ expansions do not show severe toxicity until the patient reaches 40 years of age [9,53]. Similarly, yeast cells do not display immediate cell death upon polyQ expression. However, the expression of mutant huntingtin in yeast causes severe growth deficits. Therefore, yeast can be used as an excellent cytotoxicity model for neurodegenerative diseases [47,54,55]. First, we examined whether the loss of *BLM10* affects the viability of yeast cells expressing Htt103Q, an exon1 fragment of huntingtin containing a stretch of 103 glutamines. We induced the expression of the Htt103Q and determined yeast viability by a serial dilution assay. The absence of *BLM10* caused increased cell lethality (Figure 1A, upper right panel) after induction of the expression of Htt103Q compared to wt or the *RPN4* deletion strain (*rpn4*Δ). Rpn4 is a proteasome related transcription factor. It acts as a transcriptional activator of several genes encoding proteasomal subunits [39]. The expression of Rpn4 is regulated by the 26S proteasome providing a negative feedback-loop through proteasomal degradation. According to a previous study, the loss of *BLM10* results in a mitochondrial respiratory deficit, increased mitochondrial oxidative stress, and hypersensitivity to death stimuli, while *rpn4*Δ cells did not show the same phenotype. Reduced proteasome activity by the deletion of *RPN4* primarily affects mitochondrial fusion. However, loss of *BLM10* increases mitochondrial fission indicating the specificity of Blm10 towards specific substrates [31]. Phenotypic analyses also demonstrate that manipulating the level of Rpn4 influences the replicative lifespan of yeast. Rpn4 stabilization leading to elevated proteasome capacity enhances the viability of cells against proteolytic stress, but does not influence cell response against oxidative stress [40]. In addition, a recent study demonstrated that the loss of PA200/Blm10 is the leading cause of the decline in proteasome activity during aging, which strengthens the idea that PA200/Blm10 might have a major role in age-related diseases. Furthermore, the deletion of *RPN4* decreases the level of Blm10 suggesting that Rpn4 partially promotes the transcription of Blm10 [56].

The viability of cells expressing the nontoxic huntingtin fragment composed of 25 glutamines was unaffected by the absence of *BLM10* (Figure 1A, lower panels). Next, we tested whether *blm10*Δ (*BLM10* deletion) cells would exhibit altered aggregation of the toxic Htt fragment via a filter retardation assay after 10, 14, and 18 h of induction. In both wild type (wt) and *blm10*Δ cells, accumulation of the aggregates increased in a time-dependent manner (Figure 1B). We performed statistical analysis of samples of 10 μg protein and normalized the values to wt cells. We observed a slight increase of chemiluminescent intensity in total lysate and the aggregates containing pellet after 18 h galactose induction of *blm10*Δ cells; however, the change was not significant (Figure 1C).

To detect soluble and insoluble fractions of the toxic Htt in wt and *BLM10*-deleted cells, we performed gradient gel analysis under native conditions (Figure 2A,B). We induced the expression of Htt103Q for 10, 14, and 18 h. The insoluble toxic Htt aggregates run in the stacking gel, while the soluble form runs in the separating gel. As Figure 2A,B demonstrates, we detected increasing Htt aggregate formation in both wt and *blm10*Δ cells in a time-dependent manner by normalizing the values to the corresponding 10 h induction (the first galactose induction time where we detected protein expression was 10 h). The trend of increasing Htt aggregate formation was more pronounced, but not significant in *blm10*Δ cells (Figure 2C). The data were also analyzed using a main effects ANOVA (Statistica V. 13.6), which analyzes the effects of multiple categorical independent variables. In this case, the categorical variables were cell fractions, time, and yeast strains. Both groups (wt and *blm10*Δ) changed significantly over time (*p* < 0.001). We also wanted to use other experimental approaches to detect and quantify N-Htt aggregates due to methodologic difficulties of aggregate detection and quantification under native conditions for immunoblotting, thus we performed live-cell fluorescence microscopy of cells expressing toxic (Figure 2E) or nontoxic (Figure 2D) versions of GFP-fused N-Htt. The nontoxic N-Htt with 25Q is evenly distributed in the cytosol in each cell line (Figure 2D). The loss of *BLM10* resulted in the appearance of larger Htt103Q aggregates compared to control and to *rpn4*Δ cells. (Figure 2E). Statistical analysis also indicates significantly higher number of *blm10*Δ cells with larger aggregates compared to control and to *rpn4*Δ cells (Figure 2F,G).

### 3.2. Endogenous PA200 Colocalizes with Wild Type (wt) and Mutant N-Htt in Human SH SY5Y Cells

To determine if PA200, the human orthologue of Blm10 has any physiological relevance in Huntington’s disease, we transiently transfected and overexpressed the wt and toxic N-Htt in human neuroblastoma cells and performed immunostaining. SH-SY5Y cells were transiently transfected for 48 h with constructs comprising the N-terminus of huntingtin [41]. A previous study reported that PA200 is mainly localized in the nucleus in HeLa cells and that less than 20% of PA200 is found in the cytoplasts [26]. Our immunostaining with anti-PA200 antibody in SHSY5Y cells shows both nuclear and cytoplasmic distribution of the endogenous PA200 (Figure 3A,B, left panels and Appendix A). To detect both the wt and toxic N-Htt, we used an anti-HA antibody, since a hemagglutinin (HA) tag is fused to the N-terminus. The huntingtin fragment expressing the huntingtin partial exon 1 with a normal length of 23 CAG repeats was evenly distributed in the cytosol, while the mutant with 74 CAG repeats formed nuclear and/or cytoplasmic inclusions (Figure 3A,C, middle panels). We merged the confocal images (Figure 3A,C) (superposition) and used a previously developed plug-in tool (JaCop) to the public domain ImageJ software to analyze colocalization (scatterplots). JaCop combines colocalization methods, including Pearson’s coefficient (PCC) and Manders’ overlap coefficient (MOC) [44]. Colocalization can be looked at as co-occurrence, i.e., the simple spatial overlap of two probes. As well as correlation, i.e., when the two probes not only overlap but they codistribute in proportion within and between structures [57]. The distribution of two probes are expected to overlap but not necessarily in proportion. The values we obtained for PCC (linear regression between two continuous variables) are r = 0.659 ± 0.093 for PA200 and the normal N-Htt (Figure 3B) and r = 0.72 ± 0.097 for PA200 and the toxic N-Htt (Figure 3D) indicating a moderately (<0.7) and strongly (>0.7) positive correlation between the two proteins [58,59]. MCC was also calculated using JaCoP. MCC is very sensitive to noise; therefore, to calculate M1 and M2, we set a threshold to the estimated background. M1 (or M2) shows the proportion of the green signal concurrent with the signal in the red channel over its total intensities. M1 indicates the fraction of N-HttQs-HA-tag overlapping PA200, and M2 indicates the fraction of PA200 overlapping N-HttQs-HA-tag. The M1 value of N-HttQ23-HA and PA200 is 0.35, which means that the two pixels overlap by 35%. The M2 value of PA200-N-HttQ23-HA is 0.34, which means that the two pixels overlap by 34%. (Figure 3B). For N-HttQ74-HA-PA200 (M1) and PA200- N-HttQ74-HA (M2), the two pixels overlap by 30% and 44.8%, respectively (Figure 3D) [57,60].

We previously established an SH-SY5Y cell-line to stably deplete *PMSE4* (PA200) using shRNA lentiviral technology (shPA200 cells) [35]. We validated again the knockdown efficiency by qRT-PCR and Western blotting (Appendix A). To confirm the specificity of the PA200 antibody, we tested the antibody in the shPA200 cells and corresponding controls. As Appendix A shows, the antibody is specific to PA200 and is suitable for immunofluorescent studies.

### 3.3. PA200 Interacts with wt and Mutant N-Htt In Vitro

To determine if PA200 interacts with the N-Htt-fragments, pull-down assays were performed to check the ability of the recombinant GST-Htt18Q and GST-Htt51Q to bind endogenous PA200 from human neuroblastoma cell lysates. Plasmids expressing the GST–N-huntingtin protein fusion constructs contained either 18 polyQ (control) or 51 polyQ (toxic). These constructs were modified by the addition of a TEV protease cleavage site between the GST and huntingtin protein exon 1 coding regions. The presence of the GST tag maintains both the control and the toxic variants in a soluble form. Upon cleavage of the GST moiety with TEV protease, the protein with the 51 polyQ length fragment rapidly forms high molecular weight aggregates [23,42].

GST-Htt18Q and GST-Htt51Q were successfully expressed and purified on glutathione Sepharose 4B beads. Samples resolved by SDS-PAGE were stained with Coomassie Blue dye solution and the patterns of protein bands were compared (Figure 4A). A band with an approximate size of 45 kDa was present in the GST-Htt18Q sample and a band with an approximate size of 55 kDa was present in the GST-Htt51Q. SH SY5Y cell lysates were incubated with GST or GST-fused huntingtin Q18 and Q51 proteins coupled to glutathione Sepharose beads. The interaction between the huntingtin protein fragments and PA200 was confirmed by Western blot analysis of the GST-Htt18Q and GST-Htt51Q using pull-down samples (Figure 4B and Appendix A). We also checked if we can detect the presence of the β_1_ subunit of the 20S CP using this experimental setting, but we did not detect interaction between the huntingtin protein fragments and the β_1_ subunit of the 20S CP (Figure 4C).

### 3.4. Loss of Mammalian PA200 Results in Increased Toxic N-Htt Aggregate Size and Number in Human Neuroblastoma Cells

To explore the potential role of PA200 (gene name: *PMSE4*) on the formation of mutant N-Htt aggregates, we used the previously established stable knockdown cell line (shPA200) and its respective control, stably expressing the pGIPZ-GFP [35]. We also validated the overexpression of the wt (HA-23Q) and mutant (HA-74Q) N-Htt by Western blot in both the control and shPA200 cell lines (Appendix A). Confocal microscopy images illustrate that the expression of the N-Htt fragment with a nontoxic number of polyQ is distributed normally in the cytosol of shPA200 cells similar to control cells (Figure 5A). We also analyzed the expression of mutant N-Htt in both control and shPA200 cells (Figure 5B) and found that stable loss of PA200 leads to a significant increase in mutant huntingtin aggregate size and number (Figure 5C,D).

We performed filter trap assays to analyze SDS-insoluble aggregates of the overexpressed N-Htt protein fragments and to confirm our confocal microscopy analysis. After 48 h transfection with vectors containing the wt or mutant N-terminal huntingtin, cell lysates were fractionated into supernatant (S) and pellet fraction with aggregates (P) by centrifugation. We detected a significantly higher level of SDS-insoluble aggregates of mutant huntingtin in the pellet fraction of shPA200 cells as demonstrated in Figure 5E,F. Moreover, when stained with the anti-PA200 antibody, we detected PA200 in the aggregate-containing pellet fraction of control cells, suggesting that PA200 is recruited to the mHtt-induced aggregates.

### 3.5. Blm10/PA200 Contribute to the Degradation of wt and Mutant N-Htt In Vitro

We hypothesized that Blm10/PA200 might be involved in the proteasomal degradation of wt and mutant N-Htt. Thus, we performed an in vitro assay to investigate whether Blm10-CP complexes can promote the degradation of wt huntingtin with 18Q and mutant huntingtin with 51Q glutamine stretches. For this assay, equal molar amounts of purified CP or Blm10-CP were incubated with the wt and mutant huntingtin fragments. Both CP and Blm10-CP complexes promoted the degradation of wt and mutant huntingtin fragments (Figure 6). Moreover, our data demonstrate that wt huntingtin with 18 (Figure 6B) and mutant huntingtin with 51 polyQ stretches (Figure 6A) are rapidly and significantly degraded by the proteasome in vitro in the presence of the yeast ortholog of PA200. Furthermore, we show that the wt and mutant N-Htt remain stable in the absence of the catalytically active proteasome (Figure 6C) and that the protein GST is not degraded by the proteasome (Appendix A) indicating the specificity of CP and Blm10-CP toward N-Htt. We also demonstrate that Blm10 alone does not degrade the wt or the mutant N-Htt (Appendix A). To show that Blm10 is specifically involved in the degradation of the non-ubiquitinated, soluble N-Htt Q51, we also performed the degradation assay with purified 26S. Figure 6D demonstrates that the mutant N-Htt remain stable over time in the presence of 26S. We conclude that binding of the PA200 yeast ortholog Blm10 activates the proteolytic activity of the CP, resulting in a more efficient turnover of soluble, nonaggregated huntingtin fragments.

To confirm the purity of complexes and determine if the purified CP and Blm10-CP complexes are catalytically active, we studied the purified complexes by Coomassie stain (Figure 6F left panel) and performed a native gel analysis followed by an in-gel activity assay and native gel protein stain as previously published [27,29]. CP samples were affinity purified and then subjected to size-exclusion chromatography. Blm10 degradation products are labeled with asterisk [27]. As shown in the right upper panel of Figure 6E, both purified proteasome complexes, the CP and the CP capped by Blm10, are fully active.

### 3.6. Blm10 Enhances the Ability of the Proteasome to Cut within Expanded polyQ Repeats

Previously it was published that Blm10 stimulates the trypsin-like and caspase-like protease activities, however the chymotrypsin-like activity is unaffected [29]. Thus, we set out to monitor the proteolytic processing of the recombinant GST-Htt18Q and GST-Htt51Q by mass spectrometry-based peptidomics. Recombinant, purified GST-Htt18Q and GST-Htt51Q were treated with purified CP and Blm10-CP and subjected to mass spectrometric analysis. Of the 2417 detected peptides 461 peptides were derived from the recombinant Htt constructs (Figure 7B). The peptide analysis reveals that without CP, one peptide is detectable, which originates from the Q51 and none from the Q18 Htt’s poly Q-stretch. The CP’s addition to the reaction generates a significant number of peptides from the poly-Q stretch of Q18 and Q51 Htt (second column of the heat map Figure 7A). The addition of BLM10 to the reaction generates additional peptides (e.g., SLKSFQQ) and enhances the production of other peptides also found in the CP alone reaction (e.g., KASFESLKSFQQ), as visible by an enhanced peptide intensity (third column of the heat map Figure 7A. Double cuts in the polyQ stretch are not enhanced as seen in the Q51 heat map for the poly-Q only peptides.

The addition of Blm10 to the CP leads to an increased degradation activity (Figure 7C), while the overall length distribution is not significantly changed (Figure 7D).

Next, we probed our peptide data for specific digestion patterns. Therefore, we analyzed first the differences between the identified peptides from the in vitro degradation assays. The peptides generated by the CP have a large overlap (71%) with the peptides produced by the CP-Blm10 complex (Figure 8A). The amounts of the peptides were significantly higher in the CP-Blm10 experiment (Figure 8B) indicating a higher throughput of the CP-Blm10 complex compared to the CP alone. To look deeper into changes in the amino acid sequences of the generated peptides, we analyzed the peptide sequences upstream of the cleavage sites of the identified peptides. To describe cleavage specificity, we used the Berger and Schechter nomenclature [61]. Proteolytic cleavage depends on substrate recognition by the protease [62,63]. According to Berger and Schechter, substrate residues around the protease-binding pocket are denoted as P4-P3-P2-P1↓P1′-P2′-P3′-P4′. The arrow indicates that the cleavage occurs between residues P1 and P1′ (Figure 8C). The analysis of the sequences using sequence logos (Figure 8D) and by positional enrichment of the amino acids (Figure 8E), shows the amino acid preference of the CP. The cleavage specificity of the proteasome is not significantly changed in the presence of Blm10 in the degradation assays of N-Htt fragments. The proteasome shows a preference for aliphatic and charged amino acids in positions P1, P2, and P5 (Figure 8E). The cleavage preference in position P1 is in line with the use of the chymotryptic-like CP active site. This specificity is different from the previously described stimulation of tryptic and post-glutamyl cleavage activity by PA200 [34,64] and is probably due to the use of N-Htt fragments in our study versus the use of peptides [34] or chromatin extracts [64]. The N-Htt fragments have a specific amino acid composition which will bias the use of the cleavage site.

## 4. Discussion

*S. cerevisiae* (budding yeast) has been extensively used as a valid model of protein misfolding diseases, including the neurodegenerative HD [55,65]. The advantages of studying yeast HD models include but are not limited to the following. (1) No Htt ortholog exists in yeast. Thus, the resulting phenotype, after introducing the recombinant Htt fragments in yeast, results in phenotypes independent of wild-type function. (2) Many effects of HD, such as cellular toxicity, mitochondrial dysfunction, and proteolysis, can be observed and recapitulated in a yeast model, suggesting conserved functions [43,47,54].

The role of the proteasome in the degradation of huntingtin protein is an extensively studied field. Altering or improving the activity of the proteasome by activators might offer a possible strategy to accelerate the degradation of polyQ sequences and decrease the formation of toxic aggregates. To investigate whether the degradation of N-Htt might be Blm10/PA200-dependent, we used a yeast model and validated the results in a human neuroblastoma cell line. To study cytotoxicity in yeast, we used a model developed by Meriin et al. [43]. The authors inserted the first 17 amino acids of the HTT exon-1 followed by 25 or 103 glutamines into a pYES2-vector with a galactose-inducible promoter (*GAL1* promoter). Concomitant with their published data, we showed that Htt103Q not only accumulates but also damages yeast cells.

Recently, we established that loss of *BLM10* resulted in reduced viability when cells were exposed to stress stimuli, such as acetic acid [31]. Using the inducible HD model, we studied the effects of *BLM10* loss on cellular toxicity and aggregate formation by different approaches. Our serial dilution data demonstrate that the expression of the toxic Htt103Q aggravated cellular toxicity in the *blm10*Δ strains. Interestingly, deletion of the proteasome-related transcription factor Rpn4 did not result in the same phenotype. That suggests that Blm10 might have a specific role in fighting against toxic Htt and that *blm10*Δ cells might reach the N-Htt solubility threshold earlier, leading to elevated toxicity. Furthermore, we also detected a significant increase in the number of large aggregates in the *BLM10* deletion strain compared to WT and *rpn4*Δ by confocal microscopy. We propose that the impaired degradation of N-Htt via Blm10-proteasomes enhances the formation and accumulation of toxic Htt103 aggregates and, thus, the cellular toxicity in *blm10*Δ cells.

In this study, we demonstrated that the proteasome activator Blm10/PA200 family contributes to the degradation of soluble, nonaggregated wt and mutant N-Htt in vitro. Our in vitro degradation assay indicates that both 20S CP and Blm10-CP degrade wt and toxic non-ubiquitinated N-Htt. Our results are consistent with a previously published study showing that soluble N-Htt does not inhibit 20S CP proteasomes in vitro and that the addition of CP to the reaction generates a significant number of peptides from the poly-Q stretch of Q18 and Q51 Htt [23,42,66]. To test the possibility that the proteasome activator Blm10/PA200 family might be involved in the degradation of N-Htt in vitro, we established a proteasome activity assay. We purified both 20S CP and Blm10 and purified GST-Htt18Q and GST-Htt51Q. We did not cleave the GST by TEV protease, thus, we kept the GST-Htt18Q and GST-Htt51Q in soluble forms. Our results demonstrate that the addition of purified Blm10 to the 20S CP preparation resulted in an acceleration of the degradation of non-ubiquitinated 18Q and 51Q N-Htt in vitro. Moreover, our proteomics data demonstrate that the proteasome is not clogged by the expanded polyQ repeats, but actively cleaves within polyQ stretches, and this ability is enhanced by the addition of Blm10. Thus, we propose that the Blm10/PA200 proteasome activator family targets soluble, non-ubiquitinated N-Htt polyQ repeats to proteasomal degradation and might help to delay toxic aggregate formation and might help to reach solubility threshold later in cells (Figure 9).

Here, we report for the first time that PA200, the mammalian ortholog of Blm10 shows a spatial overlap with both nontoxic wt and toxic N-Htt in human neuroblastoma cells. Moreover, we demonstrated, by pull-down assays that endogenous PA200 binds to the soluble nontoxic wt and soluble nonaggregated mutant N-Htt. However, we did not detect interaction between the catalytically active β1 subunit of 20S CP and N-Htt by our pull-down assay. This might be explained by the fact that the substrate gate is formed by the N-termini of the α-subunits, which also blocks unregulated access to the catalytic sites [67]. In addition to the results observed in yeast, we detected significantly larger aggregate size and a higher number of aggregates in human cells stably depleted of PA200. Very interestingly, PA200 is also present in mHtt aggregates, suggesting that PA200 might be recruited and accumulate in the SDS-insoluble aggregates of mHtt.

Although most of the published literature describes the mHtt aggregates as decorated by ubiquitin and that soluble mHtt is polyubiquitinated in cells transfected with mHtt, others showed that mHtt inclusion bodies (IBs) are lacking ubiquitin and that the soluble mHtt is not ubiquitinated much [23,68,69]. Here, we also describe the ubiquitin-independent degradation of soluble N-Htt fragments in vitro regulated by the Blm10/PA200 proteasomes. We speculate that the soluble wt and mutant N-Htt polyQ peptides can be degraded by the 20S CP via a ubiquitin-independent pathway. This ubiquitin-independent degradation is improved by the Blm10/PA200 activator family. The Blm10/PA200 family does not recognize ubiquitinated, globular proteins and does not participate in ATP-dependent protein degradation, but they facilitate the degradation of short peptides [29,70]. Blm10 increases the caspase-like and trypsin-like activity of the 20S CP and the recombinant PA200 increases the tryptic-like and chymotryptic-like activity of the 20S CP [29,34]. Furthermore, adding endogenous PA200 from bovine testis to purified 20S CP increases the caspase-like activity of the 20S CP and the Blm10/PA200 activator family participates in the degradation of highly unstructured protein substrates, such as tau [26,29]. In Htt, the first 17 N-terminal amino acids followed by the adjacent polyQ are predominantly highly disordered in solution [71,72]; thus, it is tempting to speculate that these highly disordered regions containing peptides are also substrates for the Blm10/PA200-proteasomes. The proteasome species exist in the cell as mixed populations. Blm10/PA200-proteasomes, for example, also exist as hybrid complexes, which can include the 19S and 20S CP, or the 20S CP as singly or doubly capped complexes, such as in the mammalian testes for PA200 and in yeast for Blm10 [30,73]. In the cellular context, the ubiquitinated Htt is recognized, unfolded, and targeted to degradation by the 26S. However, parallel to the ubiquitin-mediated pathway, the proteasome is opened up for enhanced degradation to remove mostly unfolded proteins, including Htt with higher efficiency. Thus, soluble, non-ubiquitinated polyQ monomers and oligomers might also be degraded by the Blm10/PA200-proteasome complexes. Our data serve to further elucidate the Blm10/PA200-proteasome complexes and identify new substrates. The specificities of the regulation are the subject of additional studies. Furthermore, our results highlight the value of enhancing the activity of the proteasome by proteasome activators, which may reduce toxic Htt accumulation and attenuate cellular toxicity in HD.

## Figures and Tables

**Figure 1 biomolecules-10-01581-f001:**
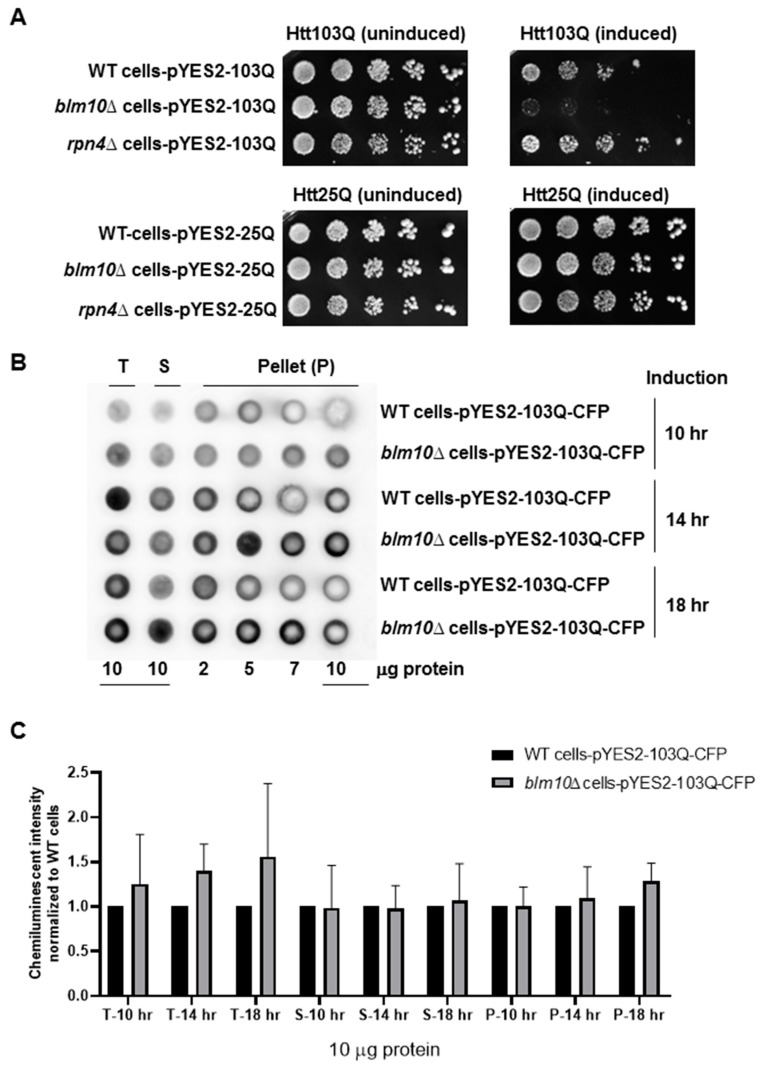
Cellular toxicity and aggregation of toxic N-terminal Htt fragments (N-Htt) in wild type (wt) and *blm10*Δ cells. (**A**) The expression of toxic huntingtin fragments impedes growth in the absence of *BLM10*. Wild type (wt), *blm10*Δ, *rpn4*Δ cells harboring Htt103Q (upper panels), and Htt25Q (lower panels were grown in galactose (induced) or raffinose-containing media (un-induced) for 18 h and spotted onto synthetic media plates. (**B**) Htt103Q aggregation in wt and *blm10*Δ cells. Expression of Htt103Q was induced by 2% galactose for the times indicated and a filter retardation assay was performed in wt and *blm10*Δ cells. The indicated protein amount of the total lysate (T), supernatant (S), and aggregate-containing pellet (P) were loaded. Htt103Q levels were detected with an anti-GFP antibody. (**C**) Quantitative analysis of SDS-insoluble aggregates in *blm10*Δ cells normalized to wt. Data are presented as mean values ± SD for four independent experiments. Data were analyzed using Image studio lite version 5.2 software. Statistical analysis was performed using ANOVA test by GraphPad Prism 8.2.1.

**Figure 2 biomolecules-10-01581-f002:**
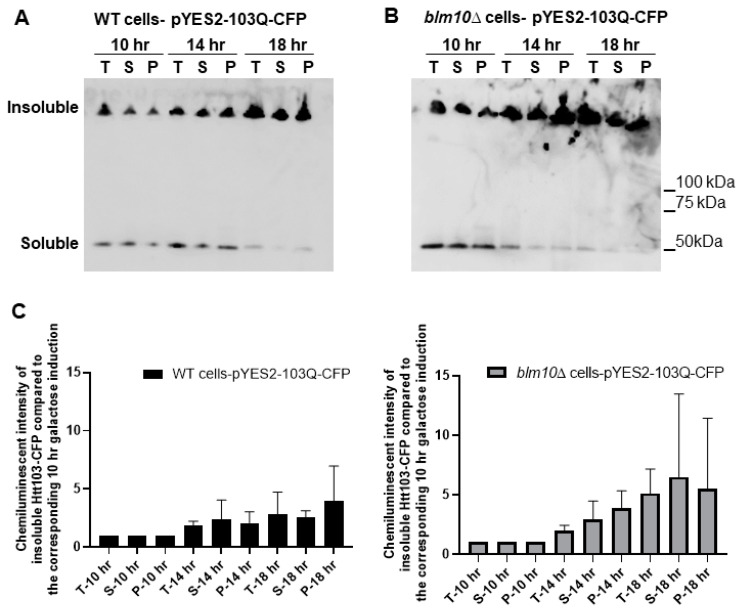
Expression and insoluble aggregate formation of toxic N-Htt in wt and *blm10*Δ cells following induction by galactose. (**A**,**B**) Expression of Htt103Q was induced by 2% galactose for the times indicated and native gel analysis was performed in wt and *blm10*Δ cells. Equal protein amounts (10 µg) of the total lysates (T), supernatants (S), and aggregate-containing pellets (P) were loaded. Htt103Q levels were detected with an anti-GFP antibody. (**C**) Quantification of theHtt103Q insoluble fractions in wt and *blm10*Δ cells were normalized to the corresponding 10 h induction in each strain. Data are shown as mean values ± SD for three independent experiments. Data were analyzed using Image Lab software version 5.2.1. Statistical analysis was performed using ANOVA by GraphPad Prism 8.2.1. *p* < 0.05 was considered significant. (**D**,**E**) Expression of toxic Htt103Q causes increased Htt103Q aggregation upon loss of *BLM10.* Visualization of N-Htt aggregation in cells expressing Htt25Q-GFP or Htt103Q-GFP using live-cell microscopy is shown. Htt103Q and Htt25Q were visualized via a GFP fusion. Differential interference contrast bright-field images (DIC) are presented to the left. Projected sequential *Z*-*stack* fluorescence images are presented. Scale bars 25 μm (inset 10 µM) (**F**,**G**) Quantification and classification of Htt103Q aggregates in wt, *blm10*Δ, and *rpn4*Δ cells. Data are shown as mean values ± SD for three independent experiments. The 409 wt, 336 *blm10*Δ, and 292 *rpn*Δ cells were counted and analyzed. Statistical analysis was performed using ANOVA by GraphPad Prism 8.2.1. (**** indicates *p* < 0.0001).

**Figure 3 biomolecules-10-01581-f003:**
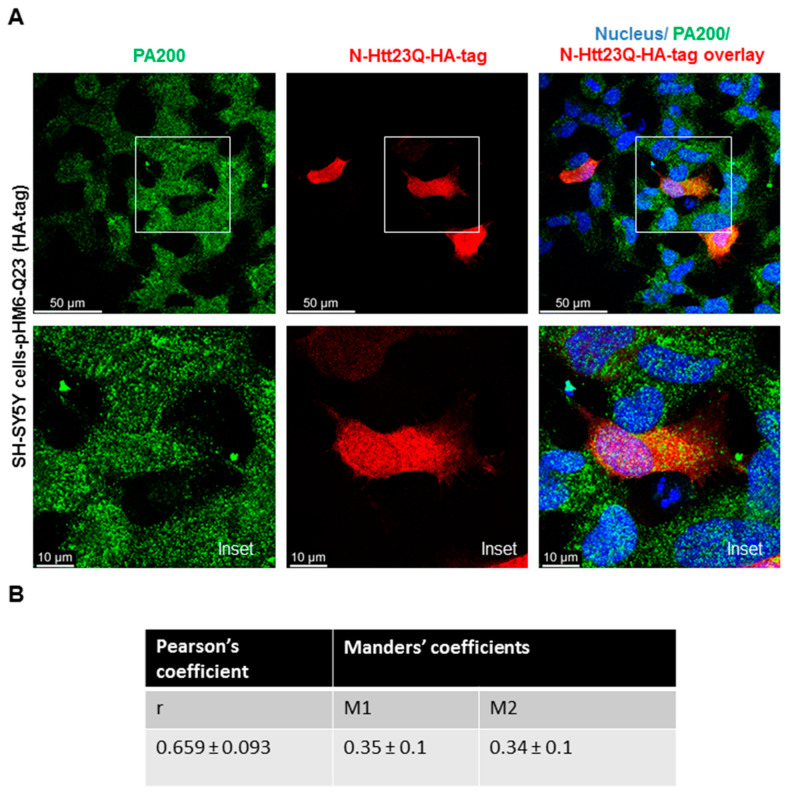
Endogenous PA200 show colocalization with N-Htt in SH-SY5Y neuroblastoma cells. Representative, merged immunofluorescence confocal images, Pearson’s and Manders’ coefficients showing colocalization of overexpressed N-Htt fragments and endogenous PA200. Cells were transfected with pHM6-Q23 and pHM6-Q74 plasmids for 48 h. Cells were immunolabeled with anti-HA-tag/Alexa Fluor 594 for Q23 and Q74 and with anti-PA200/Alexa Fluor 488 antibodies for PA200. DAPI was used to stain the cell nuclei. (**A**,**C**) The colocalization between N-Htt and PA200 is indicated by Pearson’s cross-correlation coefficient (r value) and Manders’ coefficients (M1, M2). Values were obtained by using the JACoP plugin in ImageJ. The data are presented as mean ± SD obtained from 28 images of cells expressing the N-Htt23Q with HA-tag and 24 images of cells expressing the N-Htt74Q with HA-tag. (**B**) The table shows r value between N-Htt23Q-HA-tag and PA200, M1 indicates the fraction of N-Htt23Q-HA-tag overlapping PA200, and M2 indicates the fraction of PA200 overlapping N-Htt23Q-HA-tag. (**D**) The table shows the values of r, the M1 value between N-Htt74Q-HA-tag and PA200, and the M2 value between PA200-N-Htt74Q-HA-tag.

**Figure 4 biomolecules-10-01581-f004:**
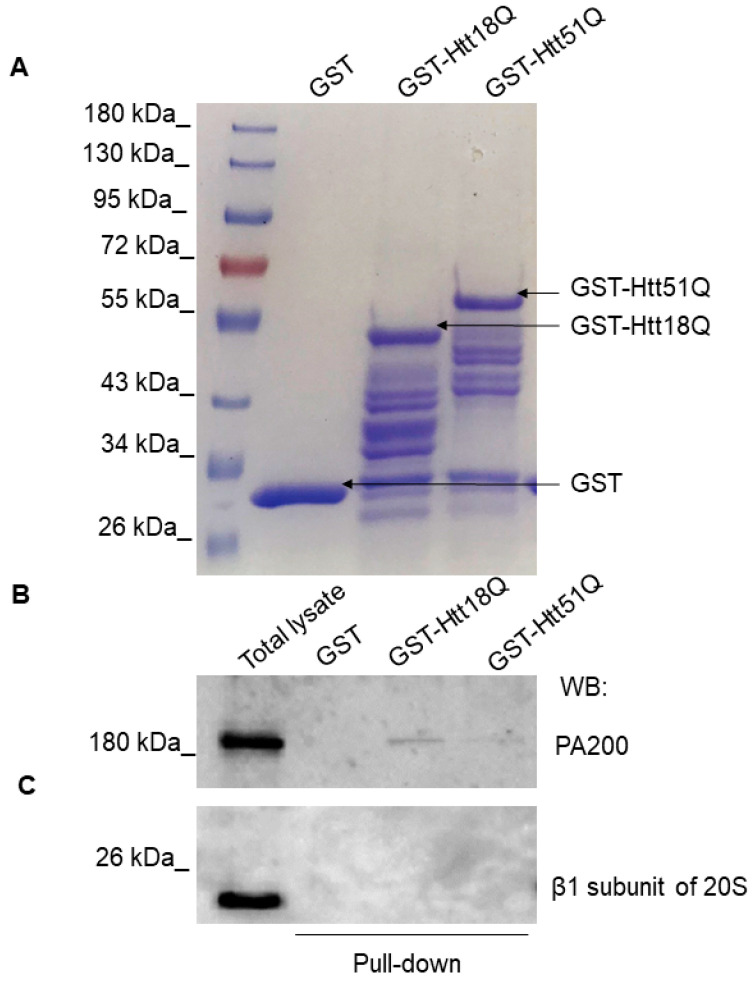
PA200 binds to wt and mutant N-Htt in vitro. (**A**) Bacterially expressed glutathione S-transferase (GST), GST-Htt18Q, and GST-Htt51Q were loaded onto glutathione sepharose and purified. The efficiency of protein purification was visualized by Coomassie Blue staining. (**B**,**C**) GST, GST-Htt18Q, and GSTHtt51Q recombinant proteins were immobilized on GSH–Sepharose 4B and incubated with SH-SY5Y neuroblastoma cell lysate. Eluted proteins were analyzed with antibodies specific for PA200 (**B**) and β_1_ subunit of 20S CP (**C**) by Western blotting.

**Figure 5 biomolecules-10-01581-f005:**
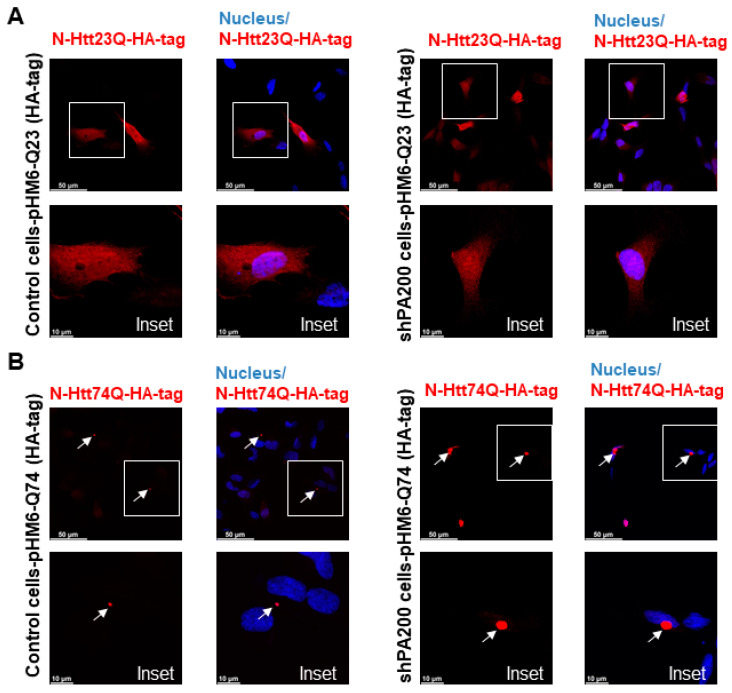
PA200 depletion leads to increased mutant N-Htt fragments aggregate size and number. (**A**,**B**) Representative confocal microscopy images of wt and mutant N-Htt fragments in control and PA200-depleted neuroblastoma human cells are shown. White squares represent the corresponding insets (lower panels); arrows indicate the toxic Htt aggregates. Control and shPA200 cells were transfected with pHM6-Q23 (with HA-tag) and pHM6-Q74 (with HA-tag) constructs for 48 h. The cells were immunolabeled with HA-tag antibody followed by Alexa Fluor 594 secondary antibody (Red). (**C**,**D**) Aggregate size and number were determined from the confocal microscopy images using ImageJ software (imagej.nih.gov). The data are presented as mean ± SD obtained from 1057 control cells and 1207 shPA200 cells from five independent experiments. Statistical analysis was carried out using unpaired Student’s *t*-test with GraphPad Prism v. 8.2.1 software (* indicates *p* < 0.05; ** indicates *p* < 0.01). (**E**) Representative images of filter trap immunoblots of SDS-resistant aggregates and PA200 protein. After 48 h transient transfection with pHM6-Q23 (HA-tag) or pHM6-Q74 (HA-tag), control and shPA200 cells were lysed with filter trap lysis buffer. Equal supernatant (S) and pellet (P) protein amounts (10 µg) from control and shPA200 cells were loaded onto 0.2 µm nitrocellulose membrane then probed with HA-tag and PA200 antibodies. Antigen–antibody complexes were detected using infrared fluorescent dye (IRDye 700 Red for HA-tag and IRDye 800 Green for PA200) secondary antibodies. Dot blots were visualized with the LI-COR Odyssey infrared imaging system. (**F**) Quantification of SDS-resistant aggregates detected by filter trap assays of control and shPA200 cells is shown. Data are shown as mean ± SD of four independent experiments. Statistical analysis was accomplished by ANOVA test using GraphPad Prism v. 8.2.1 software (*** indicates *p* < 0.001; **** indicates *p* < 0.0001).

**Figure 6 biomolecules-10-01581-f006:**
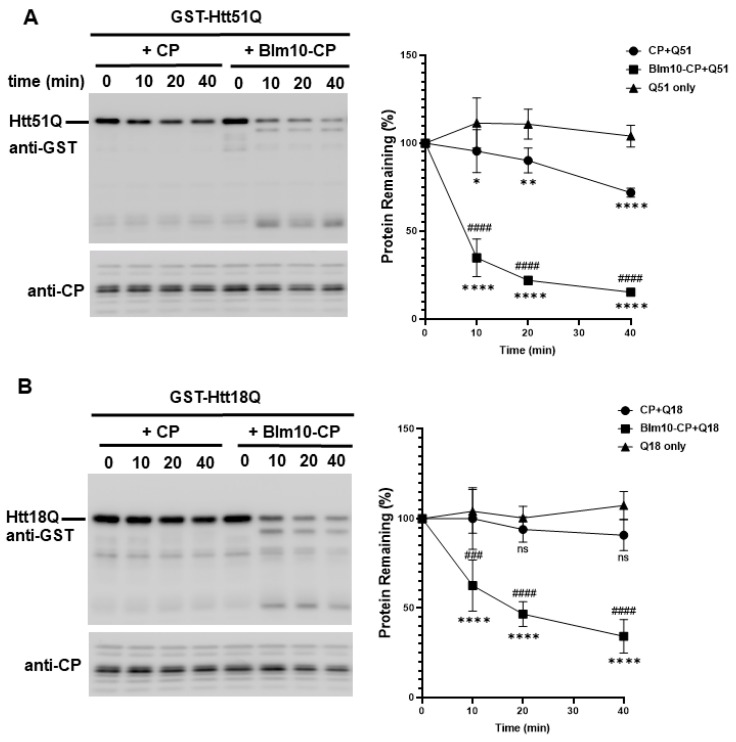
The degradation of soluble GST-Htt51Q and GST-Htt18Q is accelerated by Blm10 in vitro. (**A**,**B**) Uncleaved GST-Htt51Q and GST-Htt18Q were incubated with equal molar amounts of purified CP or Blm10-CP. (Left panels) Aliquots were taken at the times indicated and separated by SDS-PAGE. Htt51Q and Htt18Q were detected by immunoblotting with a GST-specific antibody. (Right panels) Quantification of degradation of soluble GST-Htt51Q and GST-Htt18Q by the proteasome. Data are shown as mean ± SD of three independent experiments. Statistical analysis was accomplished by ANOVA test using GraphPad Prism v. 8.2.1 software (* indicates *p* < 0.05; ** indicates *p* < 0.01; **** indicates *p* < 0.0001), * indicates statistical comparison of CP complexes with Q18 or Q51 to N-Htt samples without CP complexes, ^#^ indicates statistical comparison between CP complexes (CP and Blm10-CP) with Q18 or Q51. ^###^ indicates *p* < 0.001; ^####^ indicates *p* < 0.0001, ns indicates not significant). (**C**) GST-Htt51Q and GST-Htt18Q are not degraded and remain stable in the absence of CP. Uncleaved, soluble GST-Htt51Q and GST-Htt18Q were incubated in the absence of purified CP. Aliquots were taken at the times indicated and separated by SDS-PAGE. Htt51Q and Htt18Q were detected by immunoblotting with a GST-specific antibody. (**D**) GST-Htt51Q is not degraded by the 26S proteasome and remains stable. Uncleaved, soluble GST-Htt51Q were incubated in the presence of purified 26S. Aliquots were taken at the times indicated and separated by SDS-PAGE. Htt51Q was detected by immunoblotting with a GST-specific antibody. (**E**) (left panel) Equal molar amounts of Blm10-CP or CP were separated by SDS-PAGE and stained with Coomassie stain to demonstrate that equal amounts of CP were present in the assay. (Right panels) Blm10 degradation products are marked with an asterisk. Proteasomal complex composition (CP, Blm10-CP doubly and singly capped) was assessed by native gel electrophoresis followed by an in-gel activity assay and a native gel protein stain.

**Figure 7 biomolecules-10-01581-f007:**
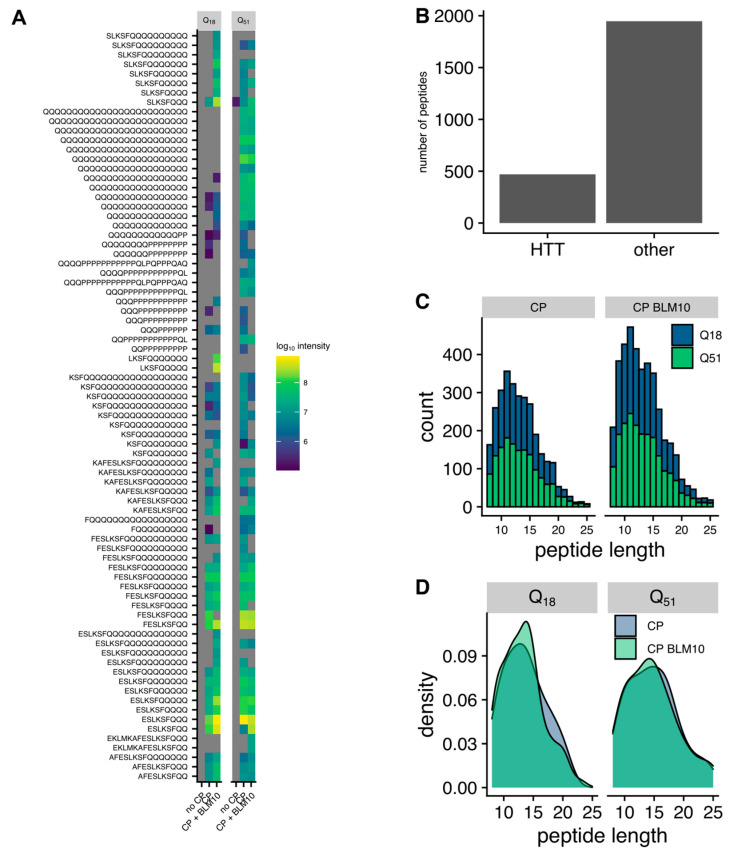
Peptidomics analysis of peptides generate by the CP and Blm10-CP complex. (**A**) Heatmap of the peptides generated in an in vitro degradation experiment of Htt18Q and Htt51Q with either no CP, CP, or the Blm10-CP complex. The color indicates the intensity of the detected peptide, while grey indicates that the peptide was not detected. (**B**) Htt derived peptides account for ≈20% of the total number of peptides detected in the in vitro reaction. (**C**) Histogram representing the peptide length distribution in the presence of the CP vs. Blm10-CP. (**D**) Density plot of the peptides generated from the degradation of Htt-18Q and Htt-51Q in the presence of CP vs. Blm10-CP.

**Figure 8 biomolecules-10-01581-f008:**
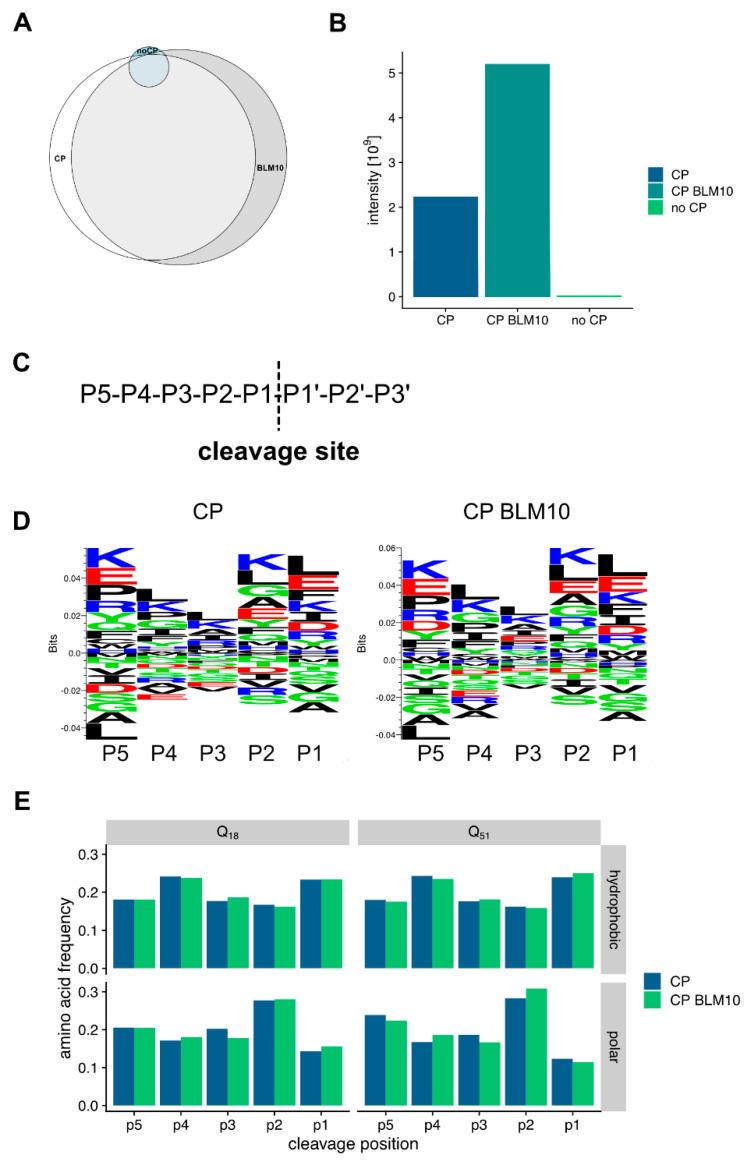
Characterization of the CP and CP BLM10 generated Htt peptides. (**A**) Venn diagram showing the overlap of peptides identified in the in vitro Htt degradation experiment. A total of 71% of the identified peptides are shared between CP and Blm10-CP based degradation. (**B**) Blm10-CP has a higher processivity for the generation of peptides. More peptides were generated at the same time from the Blm10-CP complex. (**C**) Scheme of the cleavage description. (**D**) Sequence logos of the peptides generated by the CP and Blm10-CP complex. (**E**) Amino acid frequencies for hydrophobic and polar amino acids at the different cleavage sites.

**Figure 9 biomolecules-10-01581-f009:**
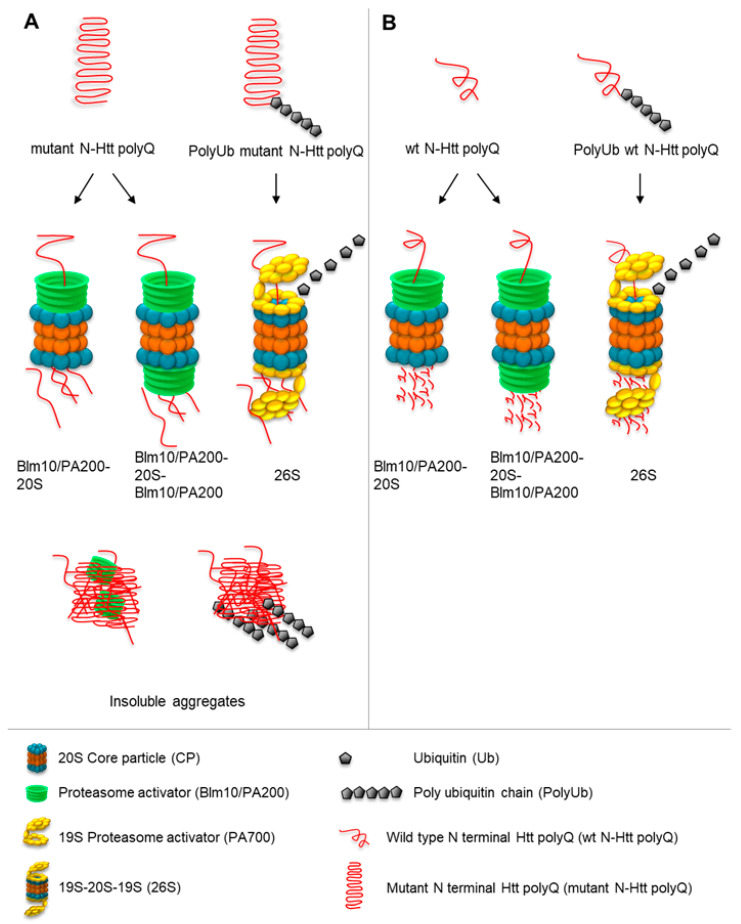
Proposed model of the involvement of Blm10/PA200 in the proteasomal degradation of mutant (**A**) and wild-type wt (**B**) N-Htt fragments. Soluble wt (**B**) and mutant (**A**) N-Htt polyQ peptides can be degraded by the Blm10/PA200-20S CP complexes via a ubiquitin-independent pathway. The ubiquitinated Htt is recognized, unfolded, and sent for degradation to the 26S proteasomes. The soluble, non-ubiquitinated N-Htt polyQ peptides are sent to Blm10/PA200-proteasomes complexes for degradation.

**Table 1 biomolecules-10-01581-t001:** Strains used in this study.

Strains	Genotypes	Ref.
BY4742	*MATα his3∆1 leu2∆0 lys2∆0 ura3∆0*	[36]
yMS122	*MATa his3 Δ1 leu2Δ0 met15Δ0 ura3Δ0 PRE1_tev_ProA (HIS3) (KanMX)GAL1pHA3BLM10*	[27]
yMS131	*MATα his3∆1 leu2∆0 met15∆0 ura3∆0 blm10∆::NatMX*	[39]
yMS268	*MATα his3∆1 leu2∆0 met15∆0 lys2∆0 ura3∆0*	[39]
yMS285	*MATα his3∆1 leu2∆0 ura3∆0 rpn4∆::KanMX*	[31]
yMS476	*MATa his3Δ1 leu2Δ0 met15Δ0 ura3Δ0 PRE1_TEV_ProA(HIS3)*	[29]
yMS1371	*MATa his3∆1 leu2∆0 met15∆0 ura3∆0+ pYES2-Htt25Q-CFP*	[40]
yMS1372	*MATa his3 ∆1 leu2Δ0 lys2Δ0 ura3Δ0 rpn4Δ::HphMX+ pYES2-Htt25Q-CFP*	This study
yMS1377	*MATa his3∆1 leu2∆0 met15∆0 ura3∆0+ pYES2-Htt103Q-CFP*	[40]
yMS1378	*MATa his3Δ1 leu2Δ0 lys2Δ0 ura3Δ0 rpn4Δ::HphMX+ pYES2-Htt103Q-CFP*	This study
yMS1392	*MATa MATα his3∆1 leu2∆0 lys2∆0 ura3∆0 blm10∆::NatMX+ pYES2-Htt25Q-CFP*	This study
yMS1393	*MATa MATα his3∆1 leu2∆0 lys2∆0 ura3∆0 blm10∆::NatMX+ pYES2-Htt103Q-CFP*	This study

**Table 2 biomolecules-10-01581-t002:** List of primary and secondary antibodies used in this study.

**Primary Antibody (Source)**	**Catalog No.**	**Host**	**Dilution**	**Method**
Anti-HA-Tag (Cell Signaling Technology, Danvers, MA, USA)	3724S	Rabbit	1:1000	Immunoblot
				Immunofluorescence
Anti-HA-Tag (Cell Signaling Technology, Danvers, MA, USA)	2367	Mouse	1:1000	Immunoblot
			1:500	Immunofluorescence
PSME4/PA200 (Novus Biologicals, Littleton, CO, USA)	NBP2-22236	Rabbit	1:2000	Immunoblot
			1:1000	Immunofluorescence
Anti-GFP (Clontech, Mountain View, CA, USA)	632592	Rabbit	1:1000	Immunoblot
Anti-GST Tag (Upstate Biotechnology, Lake Placid, NY, USA)	05-311	Mouse	1:1000	Immunoblot
β1 subunit of Proteasome 20S (human) (Enzo Biochem, Farmingdale, NY, USA).	BML-PW8140-0100	Mouse	1:1000	Immunoblot
Proteasome 20S (Yeast) core subunits (Enzo Biochem, Farmingdale, NY, USA)	BML-PW9355-0100	Rabbite	1:1000	Immunoblot
Actin (Santa Cruz Biotechnology, Dallas, TX, USA)	Sc-1616	Goat	1:10,000	Immunoblot
**Secondary Antibody (Source)**	**Catalog No.**	**Host**	**Dilution**	**Method**
Alexa fluor 594 (Thermo Fisher Scientific, Waltham, MA, USA)	A11005	Mouse	1:1000	Immunofluorescence
	A31631	Rabbit		
Alexa fluor 488 (Thermo Fisher Scientific, Waltham, MA, USA)	A11001	Mouse	1:1000	Immunofluorescence
	A11034	Rabbit		
IRDye 800CW (LI-COR, Lincoln, NE, USA)	926-32211	Rabbit	1:10,000	Immunoblot
IRDye 680RD (LI-COR, Lincoln, NE, USA)	926-68072	Mouse	1:5000	Immunoblot
Anti-Mouse IgG Antibody, HRP conjugate	A9044	Rabbit	1:3000	Immunoblot
Anti-Goat IgG Antibody, HRP conjugate	A8919	Rabbit	1:3000	Immunoblot
Anti-Rabbit IgG Antibody, HRP conjugate	A0545	Goat	1:3000	Immunoblot

**Table 3 biomolecules-10-01581-t003:** Primers used in RT-qPCR reactions.

Gene		Forward Primer	Reverse Primer
PMSE4(PA200)	Human	5′-ATGGAGAGTGCCTGAACTATTG-3′	5′-GTAGGTCAGCACACTTCCTATTC-3′
GAPDH	Human	5′-GAGTCAACGGATTTGGTCGT-3′	5′-GATCTCGCTCCTGGAAGATG-3′

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
