# Peer review of "The Proteasome Activators Blm10/PA200 Enhance the Proteasomal Degradation of N-Terminal Huntingtin"

_biomolecules, 2020, doi:10.3390/biom10111581_

Round 1

Reviewer 1 Report

There is some concern regarding the specificity of the antibody used for IF which needs to be addressed together with othe concerns. Overall an interesting study!

Author Response

Response to reviewers’ comments

We wish to express our appreciation to the Reviewers for their insightful comments, which have helped us to significantly improve the manuscript. According to the suggestions, we have thoroughly revised our manuscript and the final version is enclosed. Point-by-point responses to the comments are listed below.

Reviewer 1

“In this manuscript, Aladdin A.et al., provide in vitro and staining evidence that human PA200 binds to N-terminal Huntingtin fragments (N-Htt) and facilitates their degradation. They also observed that the aggregation of N-Htt forms and cellular toxicity increases due to loss of Blm10 in yeast strains supporting the in vitro observation that Blm10/Pa200 might be an important component of N-Htt degradation. This is a very interesting observation supported by yeast and in vitro data. There are, however, several concerns that need to be addressed.

Specific comments:

In Figure 1, it is stated that the loss of BLM10 results in a strong, time dependent accumulation of aggregates and this is nicely shown in Figure 1B. However, quantitative analysis of this is not representative. Even though the increase in T-14 and T-18 hours is more prominent than the T-10 hr in Fig. 1B, the bar graph shows more prominent increase in T-10. If the analysis is correct, it would be better to include a more representative blot in 1B.For a better understanding of the effects in blm10Δcells, values could be normalized to the WT cells.”

Response: We highly appreciate the reviewer's comment on this point. We have performed the statistical analysis again and normalized the values to the WT cells. In the first version of the figure we intended to show the formation of aggregates over time in WT cells as well as the time-dependent accumulation.

“In Figure 2, soluble and insoluble N-Htt levels were detected with protein blotting in Wt vs blm10Δcells. Quantitative analysis for the blotting is shown with a bar graph for insoluble/soluble chemiluminescence intensity. However, when compared to the Fig 2A, this graph seems weird. I would suggest plotting two separate graphs for insoluble and soluble part and normalizing the chemiluminescence intensity to the control (blm10Δcells/Wt).”

Response: We thank the reviewer for this excellent suggestion. We changed the analysis accordingly and we included the new graphs in the manuscript.

“In Fig 2C, nuclei size (Dapi staining area) for wt-103Q seems smaller than blm10Δcells with 103Q. Is this coming from different magnification? If not, what might be the reason? It would be better to add scale bars to avoid confusion. Additionally, a quantification for the cell numbers are needed as in the WT-103Q there are less cells than blm10Δ-103Q.”

Response: We strongly appreciate the comment. We went through our raw data and we included a better quality and more representative image for WT. The reviewer is right that we included an image of WT with less cells. We corrected the error. According to our raw data images we do not see smaller size nuclei. All images were taken with the same magnification, with the same microscope at the same time from the same experimental setting (The raw data is available upon request). The poorer quality of certain images might have originated from the fact that these images were taken from living yeast cells in suspension and Z-stack confocal microscopy was performed. During Z-stack processing certain cells might have moved out of focus.

We also performed statistical, quantitative analysis of WT and blm10Δ with 103QHtt, where we counted the cells numbers for both strains.

“In Figure 3, a PA200 antibody has been used. Two previous studies showedPA200 more prominently located in the nucleus than cytosol (Fabre et al., 2013; Welk et al., 2019). However, in Fig3A PA200 located mainly in the cytosol. It would be crucial yet easy to validate this antibody using the generated shPA200 cells.”

Response: We thank the reviewer’s comment. PA200 is, according to previous publications, indeed mainly localized to the cell nucleus, however it was also detected in the cytosol. We performed the immunofluorescent staining with the anti-PA200 antibody we used in this study in control and in a PA200 depleted cell line (shPA200). We included the result of the staining in the Supplementary Material as Supplementary Fig 2C The images show that we did not detect a signal for PA200 in the shPA200 cells. We also included all catalogue numbers in Table 2 for the antibodies used in this study.

In Fig. 3A, a section with more cells would be preferred to see the effect clearly (Like Fig 3B)

Response: We thank the reviewer’s comment. We replaced the panel Fig3A with more cells.

“In Figure 4, GST-Htt18Q and GST-Htt51Q were expressed and purified and used to see if there is an interaction between PA200in vitro. In Fig 4A, the expression products seem not to be pure, as there are more than one band for both products. Could this impurity affect the binding of PA200? This issue needs to be critically discussed.”

Response: We thank the reviewer’s comment. According to previously published data GST-tagged proteins might show differences following the purification protocol (PMID: 31753918). During the purification we included protease inhibitors in our setting, however we can still see on the gel protein bands running under the size of the purified protein.

The fragments are interacting with a GST antibody, thus they are most likely premature termination products of the ribosome.

We performed the western blot analysis following purification of the GST-tagged proteins for the catalytically active β1-subunit of the proteasome in mammalian cells after pull-down assay. We did not detect its interaction with the GST-tagged Htt fragments.

We speculate, that might be due to several reason:

This might be explained by the fact that the substrate gate is formed by the N-termini of the α-subunits, which also blocks unregulated access to the catalytic sites (1). PA200 might be required first to recognize the non-ubiquitinated Htt oligomers, be bound to them and PA200 might translocate the soluble oligomers to the 20S in vivo.

To strengthen our idea that Blm10/PA200 are involved in the proteasomal degradation of the non-ubiquitinated N-Htt fragments, we also performed the degradation assay with purified 26S. We included the result in Fig 6D. The result show that in vitro the 26S proteasome do not degrade the soluble, non-ubiquitinated N-Htt.

“In Figure 6, degradation kinetics of GST-Htt18Q and GST-Htt51Q are nicely shown. Impurity of the purified protein is again an issue in Fig 6C. Whether there would be an effect of this impurity needs to be discussed. The conditions that are used to produce doubly and singly capped Blm10 and CP complexes (Fig 6C) need to be detailed as this is not always straight forward.”

Response: We thank the reviewer’s comment. The high molecular bands in Fig6C are degradation products of Blm10. As the reviewer suggested we included the detailed purification protocol in the Materials and methods section. The protocol is based on a published study (2) in which the authors identified the bands by mass spectrometry as degradation products of Blm10. Degradation assays in our previous studies were performed with proteasomal complexes that were purified according to the previously mentioned purification protocol (2-4). Our group never detected side effects on the activity assays by the degradation products, as the native gel analysis followed by in gel activity assay also show full activity of the proteasomal complexes.

Purification of the singly and doubly capped Blm10-CP was performed according to the published protocol (2). We additionally included in the text the detailed purification methods. The same protocol was used to produce doubly and singly capped Blm10-CP. The doubly capped complex comes out earlier when size-exclusion column was used during purification.

“In Figure 7, it is shown that Blm10 enhances the efficiency of the 20S CP. From the Figure and the text, it not clear what samples are used for Fig 7C. From the text it seems like both GST-Htt18Q and GST-Htt51Qis used for the analysis (line 490-493). Is it the total peptide length produced from incubation of CP or CP-BLM10 withGST-Htt18Q and GST-Htt51Q? Or is it coming from either incubation with GST-Htt18Q or GST-Htt51Q?”

Response: The measurement was done separately. To improve the figure we corrected the error and recreated the figure of the results for both experiments in Figure 7.

In Fig 7D this is intuitive and can be understood easily. Another issue that needs to be addressed is the conflicting data between Fig7Dand Fig6A. In Fig 7D there seems to be no significant change of the peptide lengths when incubated with 20S CP or CP-BLM10. On the other hand, in Fig6A, there is a clear difference with CP and CP-BLM10. What could be the explanation for this difference?*In Figure 8, it is stated that there is a specific preference for amino acids at the cleavage site p2 and p4 for CP and CP-BLM10. However, in Fig 8E, the effect is not nicely seen.

Response: We thank the reviewer for pointing this out. In our model the flux through the proteasome is enhanced by the addition of the PA200. This higher flux leads to a faster degradation of the Htt-fragment proteins. The analysis of the generated peptides leads to no significant shift in the peptide length distribution as seen in the density graph 7D and to a higher number of peptides in the CP-BLM10 reaction as shown in fig 7A and C.

In Figure 8, it is stated that there is a specific preference for amino acids at the cleavage site p2 and p4 for CP and CP-BLM10. However, in Fig 8E, the effect is not nicely seen.

Response: We thank the reviewer for pointing this out. The statement in the results section has been changed from:

The analysis of the sequences using sequence logos (Fig8D) and by positional enrichment of the amino acids (Fig8E), showed that the amino acid preference of the CP, with a polar amino acid in position P4 and a hydrophobic amino acid in position P2, is preserved in the presence of Blm10 (Fig8C and D).

To:

The analysis of the sequences using sequence logos (Fig8D) and by positional enrichment of the amino acids (Fig8E), shows the amino acid preference of the CP. The cleavage specificity of proteasome is not significantly changed in the presence of the of BLM10 in the degradation assays of N-Htt fragments. The proteasome shows a preference for aliphatic and charged amino acids in positions P1, P2 and P5 (Fig8E). The cleavage preference in position P1 is in line with the use of the chymotryptic-like CP active site. This specificity is different from the previously described stimulation of tryptic and post-glutamyl clavage acitivity by PA200 (5,6) and is probably due to the use of N-Htt fragments in our study versus the use of peptides(Rego et al.) or chromatin extracts (Blickwedehl et al.). The N-Htt fragments have a specific amino acid composition which will bias the use of the cleavage side.

In Figure 9, a model is introduced for the involvement of Blm10/PA200 in the proteasomal degradation of N-Htt fragments. However, the model is confusing as there are proteasome activators (green labelled and wo 20S CP) with the ubiquitinated insoluble aggregates. This is neither mentioned nor studied in the paper and should possibly be omitted.

Response: We thank the reviewer to pointing out the error. We corrected Figure 9.

Minor comments:

We appreciate the reviewer’s minor comments. We would like to add the following to the minor coments:

  1. We shortened the Introduction part.
  2. We included the catalogue numbers of all antibodies of the study in Table 2.
  3. We included a detailed protocol for both the protein purification and degradation assay in the Materials and Methods section of the manuscript.
  4. We used several expression systems (bacterial, yeast and mammalian), therefore we used the plasmids, which are available for the scientific community and the recombinant protein can be expressed in the specific expression systems.

“Huntington’s disease is caused by the expansion (>36) of the polyglutamine (polyQ) repeat in exon 1 of the huntingtin protein. This makes the protein prone to misfolding and subsequently oligomerizing. Monomers of the mutant huntingtin protein polymerize into large aggregates or organize into different types of oligomers with different levels of cellular toxicity. “Toxic” oligomers, of a defined structure or size, are generated before the appearance of visible aggregates. (7)” It is known that higher number of polyQs are also associated with elevated toxicity and faster progression of disease in HD patients. In yeast we used an inducible system and toxicity was monitored at different time points, under the same condition for each strain. In human cells we used 74polyQ as our previously published study indicate that higher number of polyQs results in for example elevated ROS even in peripheral cells such as fibroblasts of HD patients (7).

  1. Thibaudeau, T. A., Anderson, R. T., and Smith, D. M. (2018) A common mechanism of proteasome impairment by neurodegenerative disease-associated oligomers. Nat Commun 9, 1097
  2. Schmidt, M., Haas, W., Crosas, B., Santamaria, P. G., Gygi, S. P., Walz, T., and Finley, D. (2005) The HEAT repeat protein Blm10 regulates the yeast proteasome by capping the core particle. Nat Struct Mol Biol 12, 294-303
  3. Dange, T., Smith, D., Noy, T., Rommel, P. C., Jurzitza, L., Cordero, R. J., Legendre, A., Finley, D., Goldberg, A. L., and Schmidt, M. (2011) Blm10 protein promotes proteasomal substrate turnover by an active gating mechanism. J Biol Chem 286, 42830-42839
  4. Tar, K., Dange, T., Yang, C., Yao, Y., Bulteau, A. L., Salcedo, E. F., Braigen, S., Bouillaud, F., Finley, D., and Schmidt, M. (2014) Proteasomes associated with the Blm10 activator protein antagonize mitochondrial fission through degradation of the fission protein Dnm1. J Biol Chem 289, 12145-12156
  5. Toste Rêgo, A., and da Fonseca, P. C. A. (2019) Characterization of Fully Recombinant Human 20S and 20S-PA200 Proteasome Complexes. Mol Cell 76, 138-147.e135
  6. Blickwedehl, J., Agarwal, M., Seong, C., Pandita, R. K., Melendy, T., Sung, P., Pandita, T. K., and Bangia, N. (2008) Role for proteasome activator PA200 and postglutamyl proteasome activity in genomic stability. Proc Natl Acad Sci U S A 105, 16165-16170
  7. Aladdin, A., Király, R., Boto, P., Regdon, Z., and Tar, K. (2019) Juvenile Huntington's Disease Skin Fibroblasts Respond with Elevated Parkin Level and Increased Proteasome Activity as a Potential Mechanism to Counterbalance the Pathological Consequences of Mutant Huntingtin Protein. Int J Mol Sci 20

Reviewer 2 Report

Aladdin et al. explore the role of the Blm10/PA200 proteasome core particle (CP) activator in degrading Huntingtin (Htt) with different sizes of Q expansions. Htt103Q is shown to be toxic to baker’s yeast lacking Blm10 but Htt25Q is not. The former protein also aggregates with time in blm10D relative to WT, apparently (Fig. 1C). In neuroblast transient transfectants, PA200 appears to co-aggregate with the Htt103Q aggregates, and GST-Htt fusions appear to bind to PA200 (Fig. 4). When this cell line was modified to have a stable knockdown of PA200, more Htt74Q aggregates accumulated. In vitro degradation assays with CP and CP-Blm10 complexes showed that Blm10 stimulated cleavage of GST-Htt soluble proteins (18Q and 51Q) but the generated Htt peptides were similar with and without Blm10.

Overall, this study shows that Htt is likely to be a proteolytic substrate of CP-PA200 and the yeast CP-Blm10 and that this reduces its toxicity in the yeast model. This is a modest advance, and while the conclusions overall seem likely to be correct, better data quality in places and additional quantifications and controls will be needed to make this strong enough for publication. Specific comments and questions are listed below:

Fig. 1A: Is rpn4D suppressing (slightly) the toxicity of 103Q? Any control showing that rpn4D is really deleted and impairing proteasome activity in these cells?

Are any of the differences in Fig. 1C statistically significant? Doesn't look like it. Fig. 1C is not referenced in the text, by the way.

In Fig. 2A,B at 18 h in blm10D, looks like LESS insoluble protein in stack but soluble band nearly gone; hence, still get an increase I/S ratio. These gels are very messy and are not compelling.

For Fig. 2C, Images are too small to tell if aggregates or higher levels throughout cytoplasm. Image quality is also poor.

For Fig. 3, need a more quantitative analysis. Just ONE transfected cell shown for 25Q.

Fig. 6. Why do the authors think soluble 51Q is degraded faster than 18Q by CP but the two are degraded at similar rates by CP-Blm10? This experiment also appears to have only been done once. Better in Fig. 6B would be to inactivate CP with MG132 or other CP inhibitor and show block to degradation. Current control does not rule out a contaminating protease (although not that likely).

Also, related to Fig 6. It would be interesting to determine if degradation of 51Q is blocked if the protein is allowed to aggregate (following cleavage of GST). If follow by protein staining, should be able to follow this even with Blm10-CP bands present (or can add a small epitope if needed for immunoblotting).

In Fig. 6C, left, need MW markers.

In Fig. 8, The hydrophobic vs. hydrophilic looks the opposite to me at P2 and P4 from what is stated in text. Also, what about the slight preference for hydrophobics at P1? This P1 preference would be consistent with chymotryptic-like CP active site, but Blm10 should be stimulating the other two sites relative to this site, but such a change is not obvious when comparing CP plus/minus Blm10. This is worth a comment.

Author Response

Responses to Reviewer 2:

We wish to express our appreciation to the Reviewer for the insightful comments, which have helped us to significantly improve the manuscript. According to the suggestions, we have thoroughly revised our manuscript and the final version is enclosed. Point-by-point responses to the comments are listed below.

“Aladdin et al. explore the role of the Blm10/PA200 proteasome core particle (CP) activator in degrading Huntingtin (Htt) with different sizes of Q expansions. Htt103Q is shown to be toxic to baker’s yeast lacking Blm10 but Htt25Q is not. The former protein also aggregates with time in blm10D relative to WT, apparently (Fig. 1C). In neuroblast transient transfectants, PA200 appears to co-aggregate with the Htt103Q aggregates, and GST-Htt fusions appear to bind to PA200 (Fig. 4). When this cell line was modified to have a stable knockdown of PA200, more Htt74Q aggregates accumulated. In vitro degradation assays with CP and CP-Blm10 complexes showed that Blm10 stimulated cleavage of GST-Htt soluble proteins (18Q and 51Q) but the generated Htt peptides were similar with and without Blm10.

Overall, this study shows that Htt is likely to be a proteolytic substrate of CP-PA200 and the yeast CP-Blm10 and that this reduces its toxicity in the yeast model. This is a modest advance, and while the conclusions overall seem likely to be correct, better data quality in places and additional quantifications and controls will be needed to make this strong enough for publication. Specific comments and questions are listed below:

Fig. 1A: Is rpn4D suppressing (slightly) the toxicity of 103Q? Any control showing that rpn4D is really deleted and impairing proteasome activity in these cells?

Response: We appreciate the comments.

The yMS285 rpn4Δ strain was already used in one of our previous studies (J Biol Chem. 2014 Apr 25;289(17):12145-56. doi: 10.1074/jbc.M114.554105. Epub 2014 Mar 6. PMID: 24604417). In this study, we showed that Blm10 is involved in the regulated degradation of Dnm1 (a key mitochondrial fission protein). We used the rpn4Δ strain in a cycloheximide chase assay (CHX) and we demonstrated that in the absence of RPN4, Dnm1 was stabilized over time indicating a specific role for Blm10 in the degradation of Dnm1. Furthermore, we also showed that even though Dnm1 was stabilized in rpn4Δ cells, these cells did not exhibit impaired mitochondrial function, nor do they exhibit increased oxidative damage to mitochondria.

We speculate that Blm10 has a similar role in the degradation of soluble wt or mutant N-Htt. Dnm1, similar to tau or N-Htt is a protein with highly disordered regions. We included the microscopy and the analysis of rpn4Δ cells expressing the wt and mutant N-Htt (Figure 2D-F) and we show that aggregate formation in rpn4Δ cells are similar to that of WT cells (Figure 1 also shows similar viability) however blm10Δ cells exhibit significantly higher aggregate formation compared to both WT and rpn4Δ. Thus, Blm10 delta cells might reach solubility threshold earlier with the induction time we used in our experimental setting than WT and rpn4Δ cells because in the absence of Blm10, cells are less effective to eliminate soluble mutant N-Htt, which leads to increased aggregate formation and increased toxicity.

Are any of the differences in Fig. 1C statistically significant? Doesn't look like it. Fig. 1C is not referenced in the text, by the way.

Response: We appreciate the comment. We reanalyzed the data set and normalized the effect to WT cells. We see a moderate, but unfortunately not a significant change. This might originate from the difficulties of the filter trap assay combined with the semi-quantitative character of Western blot. Often, we experienced that the aggregates practically “burnt out” the membrane even with lower amount of proteins, which makes the quantification difficult.

We included the explanation for Figure 1C.

“We observed a moderate increase of chemiluminescent intensity in total lysate and in aggregates containing pellet fraction of blm10Δ cells, however the change was not significant (Fig 1C).”

In Fig. 2A,B at 18 h in blm10D, looks like LESS insoluble protein in stack but soluble band nearly gone; hence, still get an increase I/S ratio. These gels are very messy and are not compelling.

Response: We thank the reviewer’s comment. We reanalyzed the data set again, normalized the values to WT cells after the recommendation of reviewer 1. We agree with the reviewer that the western blot does not look nice. As stated in point two that the handling of the aggregates is difficult and due to the low solubility, the separation in an SDS gel is not perfect. Even with a high degree of care, the aggregates did not focus well, while the lower soluble bands were separated fine. What we see in this experiment is a trend for soluble and insoluble N-Htts in WT and in blm10Δ cells. The size and distribution of aggregates make the quantification difficult. We include for the reviewer a result of another gradient gel analysis and if the reviewer would recommend, we will replace the image.

For Fig. 2C, Images are too small to tell if aggregates or higher levels throughout cytoplasm. Image quality is also poor.

Response: We appreciate the comment. We replaced the whole panel. We also included the results for the rpn4Δ cells. Furthermore, we included the statistical analysis of the experiments.

Of note, confocal microscopy and Z stack analysis were performed on live yeast cells. The cells were kept in small suspension drops and they were needed to be immobilized to the microscopic slide during the analysis. The fact that a few cells have moved during Z-stacking might have resulted in an “out-of focus” position in the case of a few cells. We hope that the reviewer will be satisfied with the new panel.

For Fig. 3, need a more quantitative analysis. Just ONE transfected cell shown for 25Q.

Response: We thank the reviewer’s comment. We replaced the panel for Figure 3. Furthermore, we included an analysis for co-localization using the Pearson’s coefficient to demonstrate co-localization of endogenous PA200 with both the normal and mutant N-terminal huntingtin. We also tested the specificity of the anti-PA200 antibody (Supplementary Figure 2C).

Fig. 6. Why do the authors think soluble 51Q is degraded faster than 18Q by CP but the two are degraded at similar rates by CP-Blm10? This experiment also appears to have only been done once. Better in Fig. 6B would be to inactivate CP with MG132 or other CP inhibitor and show block to degradation. Current control does not rule out a contaminating protease (although not that likely).

Response: We appreciate the comment. We performed the statistical analysis of Q51 vs Q18 degradation by the CP. The difference was not statistically significant.

We would like to apologize for the error of Figure 6. The degradation assay was performed more than three times and statistics was performed from three independent experiments. We replaced the wrong graphs with the correct graphs for the statistical analysis (Figure 6 A and B). We have also included additional controls (Supplementary Figure 5) and the degradation of Q51 by the purified 26S (Figure 6D). We also added that purified Blm10 has no catalytic activity, thus the N-Htt fragments remain stable over the degradation assay (Supplementary Figure 5).

We also would like to add the respective part of the manuscript to this section:

“Moreover, our data demonstrate that wt huntingtin with 18 (Fig. 6B) and mutant huntingtin with 51 polyQ stretches (Fig. 6A) are rapidly and significantly degraded by the proteasome in vitro in the presence of the yeast ortholog of PA200. Furthermore, we show that the wt and mutant N-Htt remain stable in the absence of the catalytically active proteasome (Fig6C) and that the protein GST is not degraded by the proteasome (Supplementary Fig.5) indicating the specificity of CP and Blm10-CP toward N-Htt. We also demonstrate that Blm10 alone does not degrade neither the wt nor the mutant N-Htt (Supplementary Fig.5). To show that Blm10 is specifically involved in the degradation of the non-ubiquitinated, soluble N-Htt Q51, we also performed the degradation assay with purified 26S. Figure 6D demonstrates that the mutant N-Htt remain stable over time in the presence of 26S. We conclude that binding of the PA200 yeast ortholog Blm10 activates the proteolytic activity of the CP, resulting in a more efficient turnover of soluble, non-aggregated huntingtin fragments.”

Also, related to Fig 6. It would be interesting to determine if degradation of 51Q is blocked if the protein is allowed to aggregate (following cleavage of GST). If follow by protein staining, should be able to follow this even with Blm10-CP bands present (or can add a small epitope if needed for immunoblotting).

Response: We appreciate the suggestion. In this study, we aim to focus on the in vitro degradation of wt and mutant N-Htt, therefore we only have preliminary data of cleaved N-Htt. Our preliminary data indicate however that the 20S used at different concentration is inhibited by the Q51 N-Htt following GST cleavage by TEV.

In Fig. 6C, left, need MW markers.

Response: We added the MW markers.

In Fig. 8, The hydrophobic vs. hydrophilic looks the opposite to me at P2 and P4 from what is stated in text. Also, what about the slight preference for hydrophobics at P1? This P1 preference would be consistent with chymotryptic-like CP active site, but Blm10 should be stimulating the other two sites relative to this site, but such a change is not obvious when comparing CP plus/minus Blm10. This is worth a comment.

Response: We thank the reviewer for pointing this out. The statement in the results section has been changed from:

The analysis of the sequences using sequence logos (Fig8D) and by positional enrichment of the amino acids (Fig8E), showed that the amino acid preference of the CP, with a polar amino acid in position P4 and a hydrophobic amino acid in position P2, is preserved in the presence of Blm10 (Fig8C and D).

To:

The analysis of the sequences using sequence logos (Fig8D) and by positional enrichment of the amino acids (Fig8E), shows the amino acid preference of the CP. The cleavage specificity of proteasome is not significantly changed in the presence of the of BLM10 in the degradation assays of N-Htt fragments. The proteasome shows a preference for aliphatic and charged amino acids in positions P1, P2 and P5 (Fig8E). The cleavage preference in position P1 is in line with the use of the chymotryptic-like CP active site. This specificity is different from the previously described stimulation of tryptic and post-glutamyl clavage acitivity by PA200 (1,2) and is probably due to the use of N-Htt fragments in our study versus the use of peptides(Rego et al.) or chromatin extracts (Blickwedehl et al.). The N-Htt fragments have a specific amino acid composition which will bias the use of the cleavage side.

  1. Toste Rêgo, A., and da Fonseca, P. C. A. (2019) Characterization of Fully Recombinant Human 20S and 20S-PA200 Proteasome Complexes. Mol Cell 76, 138-147.e135
  2. Blickwedehl, J., Agarwal, M., Seong, C., Pandita, R. K., Melendy, T., Sung, P., Pandita, T. K., and Bangia, N. (2008) Role for proteasome activator PA200 and postglutamyl proteasome activity in genomic stability. Proc Natl Acad Sci U S A 105, 16165-16170

Reviewer 3 Report

The 20S proteasome core particle (20S CP) is a multi-catalytic large protease complex highly conserved in eukaryotes. The 20S CP itself is basically latent because of its closed gates, and PA200 and its yeast ortholog Blm10 are one of the 20S CP activators. In this manuscript, the authors identified N-terminal huntingtin as a new substrate for Blm10/PA200-proteasomes. It has already been shown that Blm10 and PA200 facilitate the degradation of unstructured proteins in a ubiquitin- and ATP-independent manner, so there does not seem to be much novelty of this manuscript from the point of view of investigating novel Blm10/PA200 biological functions. However, this manuscript might be helpful to Huntington’s disease research.

  1. To compare the importance of ubiquitin-dependent and -independent degradations of Htt103Q in Fig. 1A, it would be better to examine the sensitivity to Htt103Q of some 19S mutants, such as the rpn10Δ strain.
  2. Figure 1C is not referred to and there is no description about Figure 1C in the main text.
  3. Quantification is required for the data of Htt103Q-GFP in Fig. 2C. Further, it is hard for me to see an uneven surface in the present low magnification image.
  4. It might be difficult to claim that endogenous PA200 is colocalized with N-terminal huntingtin, because endogenous PA200 exists everywhere in the SH-SY5Y cells.
  5. I cannot see any signals of nucleus in Fig. 3A. How many cells are there in this picture?
  6. Is there a difference in the binding affinities of Htt18Q and Htt51Q to PA200? The quality of blot in Fig. 4B is insufficient to know whether there is a difference or not.
  7. It might be better to clarify whether only PA200 or PA200-CP is recruited to the mHtt-induced aggregates in Fig. 5E.
  8. What are many high molecular bands in the Coomassie stained gel in Fig. 6C? Are these degradation products of Blm10?
  9. I do not understand why there is no difference in the amino acid preferences of the CP with or without Blm10, when it was published that Blm10 stimulates the trypsin-like and caspase-like protease activities. 

Author Response

Responses to Reviewer 3

We wish to express our appreciation to the Reviewers for their insightful comments, which have helped us to significantly improve the manuscript. According to the suggestions, we have thoroughly revised our manuscript and the final version is enclosed. Point-by-point responses to the comments are listed below.

The 20S proteasome core particle (20S CP) is a multi-catalytic large protease complex highly conserved in eukaryotes. The 20S CP itself is basically latent because of its closed gates, and PA200 and its yeast ortholog Blm10 are one of the 20S CP activators. In this manuscript, the authors identified N-terminal huntingtin as a new substrate for Blm10/PA200-proteasomes. It has already been shown that Blm10 and PA200 facilitate the degradation of unstructured proteins in a ubiquitin- and ATP-independent manner, so there does not seem to be much novelty of this manuscript from the point of view of investigating novel Blm10/PA200 biological functions. However, this manuscript might be helpful to Huntington’s disease research.

  1. To compare the importance of ubiquitin-dependent and -independent degradations of Htt103Q in Fig. 1A, it would be better to examine the sensitivity to Htt103Q of some 19S mutants, such as the rpn10Δ strain.

Response: We appreciate the Reviewer’s suggestion. In this study, we aim to focus on the ubiquitin independent degradation of N-Htt with the respect to the role of Blm10/PA200, and we are in the process of investigating of how the mitochondrial function is altered upon the impaired degradation of N-Htt caused by the lack of Blm10/PA200. Thus, we performed viability assays that include mitochondrial mutants. Furthermore, we show in Figure 6, that purified 26S does not promote the degradation of the soluble mutant Q51.

  1. Figure 1C is not referred to and there is no description about Figure 1C in the main text.

Response: We thank for the Reviewer’s comment. We reanalyzed the data and we included the description of Figure 1C.

“We performed statistical analysis of samples of 10 μg protein and we normalized the values to wt cells. We observed a moderate increase of chemiluminescent intensity in total lysate and in aggregates containing pellet fraction of blm10Δ cells, however the change was not significant (Fig 1C).”

  1. Quantification is required for the data of Htt103Q-GFP in Fig. 2C. Further, it is hard for me to see an uneven surface in the present low magnification image.

Response: We appreciate the comment. We replaced the whole panel for Figure 2. In addition, we also performed statistical analysis. We have also included the results of the rpn4Δ cells in the data set. We slightly modified the test and we removed the sentence describing the uneven surface of cells, because it does not contain further information.

We modified the text from:

“We also performed live-cell fluorescence microscopy of cells expressing toxic or non-toxic versions of GFP-fused Htt. The loss of BLM10 resulted in the appearance of large Htt103Q aggregates compared to control (Fig. 2C). In addition, the bright-field image (DIC) of the cells revealed an uneven surface, which we speculate is caused by the large aggregates within the cells in the absence of BLM10 (Fig. 2C).”

To:

“We also performed live-cell fluorescence microscopy of cells expressing toxic (Fig. 2E) or non-toxic (Fig. 2D) versions of GFP-fused N-Htt. The non-toxic N-Htt with 25Q is evenly distributed in the cytosol in each cell line (Fig. 2D). The loss of BLM10 resulted in the appearance of significantly larger Htt103Q aggregates compared to control and to rpn4Δ cells. (Fig. 2E).”

  1. It might be difficult to claim that endogenous PA200 is colocalized with N-terminal huntingtin, because endogenous PA200 exists everywhere in the SH-SY5Y cells.

Response: We thank the Reviewer’s comment. We performed the analysis to demonstrate the co-localization and included the description in the main text (Materials and methods, Results)

Material and methods:

“Pearson’s coefficient was evaluated using JACoP plugin in ImageJ (1). r values are Pearson’s cross-correlation coefficients indicating co-localization of N-Htt and PA200.”

Results:

“Merging of the confocal images and the high values of the Pearson’s coefficient (1) demonstrate co-localization of endogenous PA200 with both the normal and mutant N-terminal huntingtin (Fig3A and B right panels).”

  1. I cannot see any signals of nucleus in Fig. 3A. How many cells are there in this picture?

Response: We replaced the Figure 3A panel to improve the quality of the image. To analyze co-localization we used as follow:

“The co-localization between N-Htt and PA200 is indicated by the Pearson’s cross-correlation coefficient (r value). The data are presented as mean ± SD obtained from 28 images of cells expressing the N-Htt23Q with HA-tag and 24 images of cells expressing the N-Htt74Q with HA-tag.” (Figure legend for Figure 3).

For aggregate analysis, we used as follow:

“Confocal images were analyzed for aggregate number and size by ImageJ software (imagej.nih.gov). A total of 1000 cells with aggregates from each cell line were counted and analyzed.”

  1. Is there a difference in the binding affinities of Htt18Q and Htt51Q to PA200? The quality of blot in Fig. 4B is insufficient to know whether there is a difference or not.

Response: We appreciate the comment and we are planning to do a deep biochemical study in the future. We included a new panel for Figure 4. We also kept the original panel as Supplementary Figure 3. According to the figures, PA200 might have lower binding affinity to Q51 then to Q18, however this statement at this point is too speculative, therefore we did not include this statement in the main text.

  1. It might be better to clarify whether only PA200 or PA200-CP is recruited to the mHtt-induced aggregates in Fig. 5E.

Response: We thank for the comment. Pull-down assay did not detect interaction between Htt and the β1 subunit of the 20S CP alone. We included a comment on that in the discussion section and we replaced Figure 4 to show the western blot for the β1 subunit.

“However, we did not detect interaction between the catalytically active β1 subunit of 20S CP and N-Htt in the absence of PA200 by our pull-down assay. This might be explained by the fact that the substrate gate is formed by the N-termini of the α-subunits, which also blocks unregulated access to the catalytic sites (2).”

We also assayed the aggregates in our filter trap assay with the same antibody and we were not able to detect the protein.

  1. What are many high molecular bands in the Coomassie stained gel in Fig. 6C? Are these degradation products of Blm10?

Response: We thank for the comment. We replaced the panel in Figure 6 with the respective MW markers.

The high molecular bands in Fig6C are degradation products of Blm10. As Reviewer 1 suggested we included the detailed purification protocol in the Materials and methods section. The protocol is based on a published study in which the authors identified the bands by mass spectrometry as degradation products of Blm10 (3).

  1. I do not understand why there is no difference in the amino acid preferences of the CP with or without Blm10, when it was published that Blm10 stimulates the trypsin-like and caspase-like protease activities. 

Response: We appreciate the comment. The pattern we find is biased as we look only at the amino acids of the Htt protein. The pattern we find here is specific to Htt, as Htt has not an equal amino acid usage. The main reason we show the pattern is that even though the Htt contains many Qs, which should be cut by the peptidyl-glutamyl, the proteasome still shows a preference for aliphatic, charged amino acids in position P1, P2, and P5.

The pattern has no claim to be a general recognition pattern - it is specific for Htt and probably for other poly-Q disease proteins.

References

  1. Bolte, S., and Cordelières, F. P. (2006) A guided tour into subcellular colocalization analysis in light microscopy. Journal of microscopy 224, 213-232
  2. Thibaudeau, T. A., Anderson, R. T., and Smith, D. M. (2018) A common mechanism of proteasome impairment by neurodegenerative disease-associated oligomers. Nat Commun 9, 1097
  3. Schmidt, M., Haas, W., Crosas, B., Santamaria, P. G., Gygi, S. P., Walz, T., and Finley, D. (2005) The HEAT repeat protein Blm10 regulates the yeast proteasome by capping the core particle. Nat Struct Mol Biol 12, 294-303

Round 2

Reviewer 1 Report

The authors have fully adressed my concerns. The paper has considerably improved upon revision.

Author Response

Comment: The authors have fully addressed my concerns. The paper has considerably improved upon revision.

Response: We thank the referee for her/his interest in our work and for helpful comments that greatly improved the manuscript.

Reviewer 2 Report

I think the authors did a generally good job addressing my concerns. They replaced some of the figures I thought were inadequate with better data, and they do note now that some of the differences noted previously are statistically insignificant. The difficulty (almost lack of utility) of the gel-based aggregation assays is also noted. I’m not sure they have ruled out the importance of hybrid Blm10-CP-RP complexes in vivo (could have tested an RP mutant in yeast), but in vitro at least the RP does not seem to be needed.

There are still a few issues with some of the data, but I think overall, I am pretty convinced of the central claim of the paper is correct, that is, that CP-Blm10 is the major activity responsible for degrading expanded Q-repeats forms of Htt in both yeast and mammalian cells.

Fig. 2D. The DIC image suggests that the many of the rpn4D cells are dead, including the ones highlighted. Might want to get a healthier culture to analyze.

Note that my copy of manuscript had what appeared to be multiple versions of the same figures, with some truncated; was a little confusing!

Author Response

Comment: I think the authors did a generally good job addressing my concerns. They replaced some of the figures I thought were inadequate with better data, and they do note now that some of the differences noted previously are statistically insignificant. The difficulty (almost lack of utility) of the gel-based aggregation assays is also noted.

I’m not sure they have ruled out the importance of hybrid Blm10-CP-RP complexes in vivo (could have tested an RP mutant in yeast), but in vitro at least the RP does not seem to be needed.

Response: We thank the referee for her/his interest in our work and for the helpful comments that greatly improved the manuscript. We have tried to do our best to respond to the points raised. The referee has brought up some very good points and we appreciate the opportunity to clarify our research objectives and results.

We have not ruled out the importance of hybrid complexes. Our in vitro studies show that RP is not involved in the degradation of the non-ubiquitinated, soluble substrate, which agrees with the current literature. We do not rule out other possible scenarios for the degradation of the soluble Htt. For example the 26S and possibly the hybrid-complex might be involved in the degradation of the ubiquitinated Htt degradation by removing the ubiquitin-chain and cleaving the Htt into smaller peptides. Then, the Blm10/PA200-20S CP would conduct the degradation of these small peptides. We think that this argument based on our current data is too speculative and would need more experiments, which would go beyond the scope of this study.

Comment: Fig. 2D. The DIC image suggests that the many of the rpn4D cells are dead, including the ones highlighted. Might want to get a healthier culture to analyze.

Response: We appreciate the remark. We reevaluated the figure and we replaced the panel for rpn4Δ in Figure 2D.

Comment: Note that my copy of manuscript had what appeared to be multiple versions of the same figures, with some truncated; was a little confusing!

Response: We are very sorry about that. The possible reason might be is that we need to use the “Track and change option” to indicate every change we made in the manuscript. If the referee received a pdf version of the revised manuscript converted from word, the pdf version shows every single change, but also keeps the original.

Reviewer 3 Report

The revised manuscript by Aladdin and colleagues is improved over the original manuscript. It is obvious that the authors made an effort to address the criticisms that were made during the first review. However, I am not fully satisfied with some responses to my comments.

Reply to response 1

This reviewer would like the authors to describe the known phenotypes of the rpn4Δ strain and to discuss why the rpn4Δ strain did not have the same phenotypes with the blm10Δ strain in more detail.

Reply to response 4

This reviewer is not sure whether the authors’ claim is reasonable or not. It is natural for me to get high values of the Pearson’s coefficient, because PA200 and Htt23Q-HA are distributed everywhere in cells. Further, the distribution pattern of Htt74Q-HA seems apparently different from that of PA200.

Author Response

We appreciate the helpful comments of the referee. We answered her/his valuable comments in order to improve the manuscript.

Comments: This reviewer would like the authors to describe the known phenotypes of the rpn4Δ strain and to discuss why the rpn4Δ strain did not have the same phenotypes with the blm10Δ strain in more detail.

Response: We included in the body of the manucript (Section: Results 3.1) the description of the phenotype of the rpn4Δ strain as follow:

„Rpn4 is a proteasome related transcription factor. It acts as a transcriptional activator of several genes encoding proteasomal subunits (1). The expression of Rpn4 is regulated by the 26S proteasome providing a negative feedback-loop through proteasomal degradation. According to a previous study, the loss of BLM10 results in a mitochondrial respiratory deficit, increased mitochondrial oxidative stress, and hypersensitivity to death stimuli, while rpn4Δ cells did not show the same phenotype. Reduced proteasome activity by the deletion of RPN4 primarily affects mitochondrial fusion. However loss of BLM10 increases mitochondrial fission indicating the specificity of Blm10 towards specific substrates (2). Phenotypic analyses also demonstrate that manipulating the level of Rpn4 influences the replicative lifespan of yeast. Rpn4 stabilization leading to elevated proteasome capacity, enhances the viability of cells against proteolytic stress, but does not influence cell response against oxidative stress (3). In addition, a recent study demonstrated that the loss of PA200/Blm10 is the leading cause of the decline in proteasome activity during aging, which strengthen the idea that PA200/Blm10 might have a major role in age-related diseases. Furthermore, the deletion of RPN4 decreases the level of Blm10 suggesting that Rpn4 partially promotes the transcription of Blm10 (4).”

Comment: This reviewer is not sure whether the authors’ claim is reasonable or not. It is natural for me to get high values of the Pearson’s coefficient, because PA200 and Htt23Q-HA are distributed everywhere in cells. Further, the distribution pattern of Htt74Q-HA seems apparently different from that of PA200.

Response: We appreciate the helpful comment. The correlation coefficient or r value is used to assess the degree of linear correlation between two images of the same size. The r is supposed to be a matrix, where each element reflects the measure of similarity between green and red channels. The examination of complex overlays through the regions of interest (in the images) is defined by r. For example, element PA200, N-Htt23Q-HAtag shows the correlation between the images and r defines the quality of the linear relationship (corresponding offset pixels of green and red). The same for element PA200, N-Htt74Q-HAtag in which the r value was calculated in the area for red pixels and for their corresponding green pixels only (5,6).

Furthermore, previous publications demonstrate that diffuse cytosolic distribution of two proteins does not correlate with co-localization and with high values of Pearson’s coefficient, and that overlay does not necessarily mean co-localization, and investigating only overlaps might introduce bias. This is also a reason why we highly appreciate the Referee’s comment in the previous round of revision, which allowed us to apply the published method to determine co-localization based on calculating the Pearson’s coefficient (7,8).

  1. Lopez, A. D., Tar, K., Krügel, U., Dange, T., Ros, I. G., and Schmidt, M. (2011) Proteasomal degradation of Sfp1 contributes to the repression of ribosome biogenesis during starvation and is mediated by the proteasome activator Blm10. Mol Biol Cell 22, 528-540
  2. Tar, K., Dange, T., Yang, C., Yao, Y., Bulteau, A. L., Salcedo, E. F., Braigen, S., Bouillaud, F., Finley, D., and Schmidt, M. (2014) Proteasomes associated with the Blm10 activator protein antagonize mitochondrial fission through degradation of the fission protein Dnm1. J Biol Chem 289, 12145-12156
  3. Kruegel, U., Robison, B., Dange, T., Kahlert, G., Delaney, J. R., Kotireddy, S., Tsuchiya, M., Tsuchiyama, S., Murakami, C. J., Schleit, J., Sutphin, G., Carr, D., Tar, K., Dittmar, G., Kaeberlein, M., Kennedy, B. K., and Schmidt, M. (2011) Elevated proteasome capacity extends replicative lifespan in Saccharomyces cerevisiae. PLoS Genet 7, e1002253
  4. Chen, L. B., Ma, S., Jiang, T. X., and Qiu, X. B. (2020) Transcriptional upregulation of proteasome activator Blm10 antagonizes cellular aging. Biochem Biophys Res Commun 532, 211-218
  5. Bolte, S., and Cordelières, F. P. (2006) A guided tour into subcellular colocalization analysis in light microscopy. J Microsc 224, 213-232
  6. Mohapatra, S., and Weisshaar, J. C. (2018) Modified Pearson correlation coefficient for two-color imaging in spherocylindrical cells. BMC Bioinformatics 19, 428
  7. Boratkó, A., Gergely, P., and Csortos, C. (2012) Cell cycle dependent association of EBP50 with protein phosphatase 2A in endothelial cells. PLoS One 7, e35595
  8. Dunn, K. W., Kamocka, M. M., and McDonald, J. H. (2011) A practical guide to evaluating colocalization in biological microscopy. Am J Physiol Cell Physiol 300, C723-742

Round 3

Reviewer 1 Report

see below

Author Response

Responses to the Academic Editor and Reviewers:

Comment:

Dear Colleagues,

after discussion with the reviewers, we think that your manuscript deserves to be published but there are some important concerns that need to be addressed before it can be formally accepted for publication. Please answer the remarks below.

Response:

Dear Academic Editor and Reviewers,

Thank you for giving us the opportunity to resubmit a revised version of the manuscript “The proteasome activators Blm10/PA200 enhance the proteasomal degradation of N-terminal huntingtin.” for publication in Biomolecules. We appreciate the time and effort that you and the reviewers dedicated to providing feedback on our manuscript and are grateful for the insightful comments and valuable suggestions for our manuscript. We have incorporated most of the suggestions made by the Academic editor and reviewers. Those changes are highlighted within the manuscript. Please see below, for a point-by-point response to the Academic Editor’s comments and concerns.

Comment 1:

After adding statistical tests on the request of the reviewers, the authors noted that the changes in some analyses are not significant. However, they did not soften their conclusions in view of these new results. This is the case for fig 1 and 2, for which the conclusion (reused several times later in the manuscript) that 'loss of Blm10 favors aggregation' seems too strong in view of the actual data. What plays in favor of this conclusion is the convergence of the overall trend in different experiments (cell growth, aggregates etc) and not univocal experiments. It is therefore important that the authors soften this type of conclusion, both when describing the results and when reused later in the text.

According to the stats fig1B does is not representative of the results and should be changed.

Response 1: Thank you for pointing this out. The editor is correct and we have replaced Fig1B. Furthermore, accordingly, throughout the manuscript, we have refined the text to improve it and to soften the conclusion. We made changes in the text as follows from:

“In wild type (wt) cells, accumulation of the fragments was marginal, but the loss of BLM10 resulted in a time-dependent accumulation of aggregates (Fig. 1B). We performed statistical analysis of samples of 10 μg protein and we normalized the values to wt cells. We observed a moderate increase of chemiluminescent intensity in total lysate and in aggregates containing pellet fraction of blm10Δ cells, however the change was not significant (Fig. 1C).”

To:

“In both wild type (wt) and blm10Δ cells, accumulation of the aggregates increased in a time-dependent manner (Fig. 1B). We performed statistical analysis of samples of 10 μg protein and normalized the values to wt cells. We observed a slight increase of chemiluminescent intensity in total lysate and the aggregates containing pellet after 18 h galactose induction of blm10Δ cells; however, the change was not significant (Fig. 1C).”

In the discussion part, we modified the text from:

“Using the inducible HD model, we studied the effects of BLM10 loss on cellular toxicity and aggregate formation by serial dilution, filter-trap, and gradient-gel assays, respectively. Our data demonstrate that expression of the toxic Htt103Q aggravated cellular toxicity in the blm10Δ strains. Interestingly, deletion of the proteasome-related transcription factor Rpn4 did not result in the same phenotype. That suggests that Blm10 might have a specific role in fighting against toxic Htt and that blm10Δ cells might reach the N-Htt solubility threshold earlier leading to elevated toxicity. Furthermore, we also detected a moderate increase in the number of aggregates and an elevated level of insoluble aggregates in the BLM10 deletion strain compared to WT.”

To:

“Using the inducible HD model, we studied the effects of BLM10 loss on cellular toxicity and aggregate formation by different approaches. Our serial dilution data demonstrate that expression of the toxic Htt103Q aggravated cellular toxicity in the blm10Δ strains. Interestingly, deletion of the proteasome-related transcription factor Rpn4 did not result in the same phenotype. That suggests that Blm10 might have a specific role in fighting against toxic Htt and that blm10Δ cells might reach the N-Htt solubility threshold earlier, leading to elevated toxicity. Furthermore, we also detected a significant increase in the number of large aggregates in the BLM10 deletion strain compared to WT and rpn4Δ by confocal microscopy.”

We also changed the title of Figure 1 from: Figure 1. Elevated toxicity and aggregation of toxic N-Htt in the absence of BLM10.

To:  Figure 1. Cellular toxicity and aggregation of toxic N-Htt in wt and blm10Δ cells.

and Figure 2 from: Figure 2. Insoluble fractions of toxic N-Htt are elevated and soluble fractions are reduced in blm10D cells in a time-dependent manner after induction.

To:

Figure 2. Expression and insoluble aggregate formation of toxic N-Htt in wt and blm10Δ cells following induction by galactose.

Comment 2:

In figure 2, it is not appropriate to quantitatively compare signals from different blots, as there is no guaranty that the transfer and/or the development of the blots have been done with the same efficiency. To compare wt vs mutant cells, all values should be normalized against a control signal present in each blot, or the gels should be redone such as compared wt and mutant samples are analyzed in the same gel and blot.

Response 2:

We agree that we cannot compare signals from different blots quantitatively, even though the loading of samples for native gel analysis and the transfer of proteins were all performed in the same chamber under the same condition at the same time. The development of blots was also performed under the same condition at the same time for the same exposure time. As control, we used the zero time point Galactose induction, but this time point does not give any signal for the Htt proteins, because the proteins do not start to express yet. In our experimental setting, the 10 hr induction of Htt expression provided the first detectable signal. Therefore, we reevaluated our analysis and we performed the analysis separately for each strain samples (wt and Blm10Δ). We compared the expression of proteins of different induction times to 10 hr induction in each sample. Similar to the previous statistical analysis, we have unfortunately not obtained significant changes. Therefore, we have modified the manuscript and softened our conclusion. What we would like to point out is that even though we do not see significant changes (as we explained in Round1 revision, we feel that this is due to the methodologic challenge of this experiment), we see a trend of moderate increase of aggregate formation in blm10Δ cells.

The text was modified from:

“As Figures 2A and C demonstrate, loss of BLM10 resulted in higher aggregate formation in a time-dependent manner compared to wt cells, although the change was not significant. Of note, the soluble, non-aggregated mHtt protein level was reduced after 14 h of induction upon loss of BLM10, while the soluble mHtt protein was reduced in wt cells only after 18 h of induction (Fig. 2B). We also performed live-cell fluorescence microscopy of cells expressing toxic (Fig. 2E) or non-toxic (Fig. 2D) versions of GFP-fused N-Htt. The non-toxic N-Htt with 25Q is evenly distributed in the cytosol in each cell line (Fig. 2D).”

To:

“As Figures 2A and B demonstrate, we detected increasing Htt aggregate formation in both wt and blm10Δ cells in a time-dependent manner by normalizing the values to the corresponding 10 h induction (the first galactose induction time where we detected protein expression was 10 h). The trend of increasing Htt aggregate formation was more pronounced, but not significant in blm10Δ cells (Fig. 2C). The data were also analyzed using a main effects ANOVA (Statistica V. 13.6), which analyzes the effects of multiple categorical independent variables. In this case, the categorical variables were cell fractions, time, and yeast strains. Both groups (wt and blm10Δ) changed significantly over time (p<0.001). We also wanted to use other experimental approaches to detect and quantify N-Htt aggregates due to methodologic difficulties of aggregate detection and quantification under native conditions for immunoblotting, thus we performed live-cell fluorescence microscopy of cells expressing toxic (Fig. 2E) or non-toxic (Fig. 2D) versions of GFP-fused N-Htt.”

Comment 3:

Regarding the cellular localization of PA200, it is not justified to claim that it “in accordance" with Ustrell et al's article (ref 26) as these authors concluded that the protein is mostly nuclear.

Response 3:

We thank the editor for the comment. We have modified the text from:

“Immunostaining with anti-PA200 antibody shows both nuclear and cytoplasmic distribution of the endogenous PA200 in accordance with previously published data (26)”

To:

“A previous study reported that PA200 is mainly localized in the nucleus in HeLa cells and that less than 20% of PA200 is found in the cytoplasts (26). Our immunostaining with anti-PA200 antibody in SHSY5Y cells shows both nuclear and cytoplasmic distribution of the endogenous PA200 (Fig. 3A and B left panels and Supplementary Fig. 1).”

Comment 4:

Also, what is the justification to call 'a high value' a Pearson coefficient of ≈0.7? Actually the use of the Pearson coefficient to justify the colocalization of PA200 with N-Htt is probably a weak argument in a context where the 2 proteins are rather evenly distributed (Fig3A) or in the opposite are distributed completely differently (Fig3B). In fact, the article cited in the manuscript to describe Pearson coefficient (ref 44) is stating very clearly that it is a not a good tool to quantify co-localization. The authors should discuss more these different issues and soften their conclusion accordingly.

Response 4: While we appreciate the editor’s feedback, we respectfully disagree. The plugin JaCop from reference 44 that we used to analyze co-localization combines several methods of co-localization including the Pearson’s coefficient. However, we have modified the text and described the analysis in more detail. We modified the text from:

“Merging of the confocal images and the high values of the Pearson’s coefficient (44) demonstrate co-localization of endogenous PA200 with both the normal and mutant N-terminal huntingtin (Fig. 3A and B right panels).”

To:

“We merged the confocal images (Fig. 3A and C) (superposition) and used a previously developed plug-in tool (JaCop) to the public domain ImageJ software to analyze co-localization (scatterplots). JaCop combines co-localization methods, including Pearson’s coefficient (PCC) and Manders’ overlap coefficient (MOC)(44). Co-localization can be looked at as co-occurrence, i.e. the simple spatial overlap of two probes. as well as correlation, i.e. when the two probes not only overlap but they co-distribute in proportion within and between structures (57). The distribution of two probes are expected to overlap but not necessarily in proportion. The values we obtained for PCC (linear regression between two continuous variables) are r= 0.659 ± 0.093 for PA200 and the normal N-Htt (Fig. 3B) and r= 0.72 ± 0.097 for PA200 and the toxic N-Htt (Fig. 3D) indicating a moderately (< 0.7) and strongly (> 0.7) positive correlation between the two proteins (58,59). MCC was also calculated using JaCoP. MCC is very sensitive to noise; therefore, to calculate M1 and M2, we set a threshold to the estimated background. M1 (or M2) shows the proportion of the green signal concurrent with the signal in the red channel over its total intensities. M1 indicates the fraction of N-HttQs-HA-tag overlapping PA200, and M2 indicates the fraction of PA200 overlapping N-HttQs-HA-tag. The M1 value of N-HttQ23-HA and PA200 is 0.35, which means that the two pixels overlap of 35%. The M2 value of PA200-N-HttQ23-HA is 0.34, which means that the two pixels overlap of 34% (Fig. 3B). For N-HttQ74-HA-PA200 (M1) and PA200- N-HttQ74-HA (M2), the two pixels overlap of 30% and 44.8%, respectively (Fig. 3D) (57,60).”

Comment 5: In Fig6, the molar ratio substrate/enzyme is less than 4 (40nM vs 12nM). The fact that the reaction is barely enzymatic should be discussed for a clear interpretation of these data.

Response 5: We thank the editor for the comment. We feel that we provided controls that demonstrate that the in vitro degradation assay was not affected by non-specific proteases. The controls include the stabilized GST protein over time in the presence of CP and Blm10-CP, showing the specificity of the proteasome toward the substrate (Suppl.Fig5). Furthermore, we show the stabilization of substrates over time in the absence of CP and Blm10-CP (Suppl. Fig5). We also provide a control experiment, which shows that the Htt substrates are stabilized when CP is missing but Blm10 is present (Suppl. Fig5). That demonstrates that the CP possesses the catalytic activity. We also demonstrate that the purified proteasome that we used in the degradation assay is catalytically active (Figure 6E).

We agree with the editor that we have not determined the Km and Vmax values; however, we have tried several concentrations of substrates, including 150, 300, and 600 nM substrate concentrations (please see the pdf version). As the additional data show, using these concentrations resulted in similar turnover efficiency; however, for detection and quantification purposes, we used the 40 nM.

Comment 6:

In figure 7A, what are the exact arguments that justify the claim that 'the ability to cut in polyQ stretches is enhanced by the addition of Blm10'? They should be made more explicit.

Response 6: We appreciate the comment. We elaborated our argument, and have modified the text accordingly from:

“The analysis of the peptides clearly shows that the CP alone is able to cut within extended polyQ stretches (Fig. 7A) concomitant with a previously published data demonstrating that mammalian proteasomes are capable of degrading expanded polyQ sequences (57). Furthermore, this ability to cut in polyQ stretches is enhanced by the addition of Blm10 especially for sequences which contain non-polyQ sequences at the N-terminal, while the C-terminal extension does not show the cleavage enhancement (Fig. 7A).”

To:

“The peptide analysis reveals that without CP, one peptide is detectable, which originates from the Q51 and none from the Q18 Htt's poly Q-stretch. The CP's addition to the reaction generates a significant number of peptides from the poly-Q stretch of Q18 and Q51 Htt (second column of the heat map Fig. 7a), which is in line with previously published data (66). The addition of BLM10 to the reaction generates additional peptides (e.g. SLKSFQQQ...) and enhances the production of other peptides also found in the CP alone reaction (e.g. KASFESLKSFQQ...), as visible by an enhanced peptide intensity (third column of the heat map Fig. 7a). Double cuts in the polyQ stretch are not enhanced, as seen in the Q51 heat map for the poly-Q only peptides.”

Reviewer 2 Report

I am supportive of publication.

Author Response

Responses to the Academic Editor and Reviewers:

Comment:

Dear Colleagues,

after discussion with the reviewers, we think that your manuscript deserves to be published but there are some important concerns that need to be addressed before it can be formally accepted for publication. Please answer the remarks below.

Response:

Dear Academic Editor and Reviewers,

Thank you for giving us the opportunity to resubmit a revised version of the manuscript “The proteasome activators Blm10/PA200 enhance the proteasomal degradation of N-terminal huntingtin.” for publication in Biomolecules. We appreciate the time and effort that you and the reviewers dedicated to providing feedback on our manuscript and are grateful for the insightful comments and valuable suggestions for our manuscript. We have incorporated most of the suggestions made by the Academic editor and reviewers. Those changes are highlighted within the manuscript. Please see below, for a point-by-point response to the Academic Editor’s comments and concerns.

Comment 1:

After adding statistical tests on the request of the reviewers, the authors noted that the changes in some analyses are not significant. However, they did not soften their conclusions in view of these new results. This is the case for fig 1 and 2, for which the conclusion (reused several times later in the manuscript) that 'loss of Blm10 favors aggregation' seems too strong in view of the actual data. What plays in favor of this conclusion is the convergence of the overall trend in different experiments (cell growth, aggregates etc) and not univocal experiments. It is therefore important that the authors soften this type of conclusion, both when describing the results and when reused later in the text.

According to the stats fig1B does is not representative of the results and should be changed.

Response 1: Thank you for pointing this out. The editor is correct and we have replaced Fig1B. Furthermore, accordingly, throughout the manuscript, we have refined the text to improve it and to soften the conclusion. We made changes in the text as follows from:

“In wild type (wt) cells, accumulation of the fragments was marginal, but the loss of BLM10 resulted in a time-dependent accumulation of aggregates (Fig. 1B). We performed statistical analysis of samples of 10 μg protein and we normalized the values to wt cells. We observed a moderate increase of chemiluminescent intensity in total lysate and in aggregates containing pellet fraction of blm10Δ cells, however the change was not significant (Fig. 1C).”

To:

“In both wild type (wt) and blm10Δ cells, accumulation of the aggregates increased in a time-dependent manner (Fig. 1B). We performed statistical analysis of samples of 10 μg protein and normalized the values to wt cells. We observed a slight increase of chemiluminescent intensity in total lysate and the aggregates containing pellet after 18 h galactose induction of blm10Δ cells; however, the change was not significant (Fig. 1C).”

In the discussion part, we modified the text from:

“Using the inducible HD model, we studied the effects of BLM10 loss on cellular toxicity and aggregate formation by serial dilution, filter-trap, and gradient-gel assays, respectively. Our data demonstrate that expression of the toxic Htt103Q aggravated cellular toxicity in the blm10Δ strains. Interestingly, deletion of the proteasome-related transcription factor Rpn4 did not result in the same phenotype. That suggests that Blm10 might have a specific role in fighting against toxic Htt and that blm10Δ cells might reach the N-Htt solubility threshold earlier leading to elevated toxicity. Furthermore, we also detected a moderate increase in the number of aggregates and an elevated level of insoluble aggregates in the BLM10 deletion strain compared to WT.”

To:

“Using the inducible HD model, we studied the effects of BLM10 loss on cellular toxicity and aggregate formation by different approaches. Our serial dilution data demonstrate that expression of the toxic Htt103Q aggravated cellular toxicity in the blm10Δ strains. Interestingly, deletion of the proteasome-related transcription factor Rpn4 did not result in the same phenotype. That suggests that Blm10 might have a specific role in fighting against toxic Htt and that blm10Δ cells might reach the N-Htt solubility threshold earlier, leading to elevated toxicity. Furthermore, we also detected a significant increase in the number of large aggregates in the BLM10 deletion strain compared to WT and rpn4Δ by confocal microscopy.”

We also changed the title of Figure 1 from: Figure 1. Elevated toxicity and aggregation of toxic N-Htt in the absence of BLM10.

To:  Figure 1. Cellular toxicity and aggregation of toxic N-Htt in wt and blm10Δ cells.

and Figure 2 from: Figure 2. Insoluble fractions of toxic N-Htt are elevated and soluble fractions are reduced in blm10D cells in a time-dependent manner after induction.

To:

Figure 2. Expression and insoluble aggregate formation of toxic N-Htt in wt and blm10Δ cells following induction by galactose.

Comment 2:

In figure 2, it is not appropriate to quantitatively compare signals from different blots, as there is no guaranty that the transfer and/or the development of the blots have been done with the same efficiency. To compare wt vs mutant cells, all values should be normalized against a control signal present in each blot, or the gels should be redone such as compared wt and mutant samples are analyzed in the same gel and blot.

Response 2:

We agree that we cannot compare signals from different blots quantitatively, even though the loading of samples for native gel analysis and the transfer of proteins were all performed in the same chamber under the same condition at the same time. The development of blots was also performed under the same condition at the same time for the same exposure time. As control, we used the zero time point Galactose induction, but this time point does not give any signal for the Htt proteins, because the proteins do not start to express yet. In our experimental setting, the 10 hr induction of Htt expression provided the first detectable signal. Therefore, we reevaluated our analysis and we performed the analysis separately for each strain samples (wt and Blm10Δ). We compared the expression of proteins of different induction times to 10 hr induction in each sample. Similar to the previous statistical analysis, we have unfortunately not obtained significant changes. Therefore, we have modified the manuscript and softened our conclusion. What we would like to point out is that even though we do not see significant changes (as we explained in Round1 revision, we feel that this is due to the methodologic challenge of this experiment), we see a trend of moderate increase of aggregate formation in blm10Δ cells.

The text was modified from:

“As Figures 2A and C demonstrate, loss of BLM10 resulted in higher aggregate formation in a time-dependent manner compared to wt cells, although the change was not significant. Of note, the soluble, non-aggregated mHtt protein level was reduced after 14 h of induction upon loss of BLM10, while the soluble mHtt protein was reduced in wt cells only after 18 h of induction (Fig. 2B). We also performed live-cell fluorescence microscopy of cells expressing toxic (Fig. 2E) or non-toxic (Fig. 2D) versions of GFP-fused N-Htt. The non-toxic N-Htt with 25Q is evenly distributed in the cytosol in each cell line (Fig. 2D).”

To:

“As Figures 2A and B demonstrate, we detected increasing Htt aggregate formation in both wt and blm10Δ cells in a time-dependent manner by normalizing the values to the corresponding 10 h induction (the first galactose induction time where we detected protein expression was 10 h). The trend of increasing Htt aggregate formation was more pronounced, but not significant in blm10Δ cells (Fig. 2C). The data were also analyzed using a main effects ANOVA (Statistica V. 13.6), which analyzes the effects of multiple categorical independent variables. In this case, the categorical variables were cell fractions, time, and yeast strains. Both groups (wt and blm10Δ) changed significantly over time (p<0.001). We also wanted to use other experimental approaches to detect and quantify N-Htt aggregates due to methodologic difficulties of aggregate detection and quantification under native conditions for immunoblotting, thus we performed live-cell fluorescence microscopy of cells expressing toxic (Fig. 2E) or non-toxic (Fig. 2D) versions of GFP-fused N-Htt.”

Comment 3:

Regarding the cellular localization of PA200, it is not justified to claim that it “in accordance" with Ustrell et al's article (ref 26) as these authors concluded that the protein is mostly nuclear.

Response 3:

We thank the editor for the comment. We have modified the text from:

“Immunostaining with anti-PA200 antibody shows both nuclear and cytoplasmic distribution of the endogenous PA200 in accordance with previously published data (26)”

To:

“A previous study reported that PA200 is mainly localized in the nucleus in HeLa cells and that less than 20% of PA200 is found in the cytoplasts (26). Our immunostaining with anti-PA200 antibody in SHSY5Y cells shows both nuclear and cytoplasmic distribution of the endogenous PA200 (Fig. 3A and B left panels and Supplementary Fig. 1).”

Comment 4:

Also, what is the justification to call 'a high value' a Pearson coefficient of ≈0.7? Actually the use of the Pearson coefficient to justify the colocalization of PA200 with N-Htt is probably a weak argument in a context where the 2 proteins are rather evenly distributed (Fig3A) or in the opposite are distributed completely differently (Fig3B). In fact, the article cited in the manuscript to describe Pearson coefficient (ref 44) is stating very clearly that it is a not a good tool to quantify co-localization. The authors should discuss more these different issues and soften their conclusion accordingly.

Response 4: While we appreciate the editor’s feedback, we respectfully disagree. The plugin JaCop from reference 44 that we used to analyze co-localization combines several methods of co-localization including the Pearson’s coefficient. However, we have modified the text and described the analysis in more detail. We modified the text from:

“Merging of the confocal images and the high values of the Pearson’s coefficient (44) demonstrate co-localization of endogenous PA200 with both the normal and mutant N-terminal huntingtin (Fig. 3A and B right panels).”

To:

“We merged the confocal images (Fig. 3A and C) (superposition) and used a previously developed plug-in tool (JaCop) to the public domain ImageJ software to analyze co-localization (scatterplots). JaCop combines co-localization methods, including Pearson’s coefficient (PCC) and Manders’ overlap coefficient (MOC)(44). Co-localization can be looked at as co-occurrence, i.e. the simple spatial overlap of two probes. as well as correlation, i.e. when the two probes not only overlap but they co-distribute in proportion within and between structures (57). The distribution of two probes are expected to overlap but not necessarily in proportion. The values we obtained for PCC (linear regression between two continuous variables) are r= 0.659 ± 0.093 for PA200 and the normal N-Htt (Fig. 3B) and r= 0.72 ± 0.097 for PA200 and the toxic N-Htt (Fig. 3D) indicating a moderately (< 0.7) and strongly (> 0.7) positive correlation between the two proteins (58,59). MCC was also calculated using JaCoP. MCC is very sensitive to noise; therefore, to calculate M1 and M2, we set a threshold to the estimated background. M1 (or M2) shows the proportion of the green signal concurrent with the signal in the red channel over its total intensities. M1 indicates the fraction of N-HttQs-HA-tag overlapping PA200, and M2 indicates the fraction of PA200 overlapping N-HttQs-HA-tag. The M1 value of N-HttQ23-HA and PA200 is 0.35, which means that the two pixels overlap of 35%. The M2 value of PA200-N-HttQ23-HA is 0.34, which means that the two pixels overlap of 34% (Fig. 3B). For N-HttQ74-HA-PA200 (M1) and PA200- N-HttQ74-HA (M2), the two pixels overlap of 30% and 44.8%, respectively (Fig. 3D) (57,60).”

Comment 5: In Fig6, the molar ratio substrate/enzyme is less than 4 (40nM vs 12nM). The fact that the reaction is barely enzymatic should be discussed for a clear interpretation of these data.

Response 5: We thank the editor for the comment. We feel that we provided controls that demonstrate that the in vitro degradation assay was not affected by non-specific proteases. The controls include the stabilized GST protein over time in the presence of CP and Blm10-CP, showing the specificity of the proteasome toward the substrate (Suppl.Fig5). Furthermore, we show the stabilization of substrates over time in the absence of CP and Blm10-CP (Suppl. Fig5). We also provide a control experiment, which shows that the Htt substrates are stabilized when CP is missing but Blm10 is present (Suppl. Fig5). That demonstrates that the CP possesses the catalytic activity. We also demonstrate that the purified proteasome that we used in the degradation assay is catalytically active (Figure 6E).

We agree with the editor that we have not determined the Km and Vmax values; however, we have tried several concentrations of substrates, including 150, 300, and 600 nM substrate concentrations (please see the pdf version of this comment). As the additional data show, using these concentrations resulted in similar turnover efficiency; however, for detection and quantification purposes, we used the 40 nM.

Comment 6:

In figure 7A, what are the exact arguments that justify the claim that 'the ability to cut in polyQ stretches is enhanced by the addition of Blm10'? They should be made more explicit.

Response 6: We appreciate the comment. We elaborated our argument, and have modified the text accordingly from:

“The analysis of the peptides clearly shows that the CP alone is able to cut within extended polyQ stretches (Fig. 7A) concomitant with a previously published data demonstrating that mammalian proteasomes are capable of degrading expanded polyQ sequences (57). Furthermore, this ability to cut in polyQ stretches is enhanced by the addition of Blm10 especially for sequences which contain non-polyQ sequences at the N-terminal, while the C-terminal extension does not show the cleavage enhancement (Fig. 7A).”

To:

“The peptide analysis reveals that without CP, one peptide is detectable, which originates from the Q51 and none from the Q18 Htt's poly Q-stretch. The CP's addition to the reaction generates a significant number of peptides from the poly-Q stretch of Q18 and Q51 Htt (second column of the heat map Fig. 7a), which is in line with previously published data (66). The addition of BLM10 to the reaction generates additional peptides (e.g. SLKSFQQQ...) and enhances the production of other peptides also found in the CP alone reaction (e.g. KASFESLKSFQQ...), as visible by an enhanced peptide intensity (third column of the heat map Fig. 7a). Double cuts in the polyQ stretch are not enhanced, as seen in the Q51 heat map for the poly-Q only peptides.”

Reviewer 3 Report

I read the comments by guest editor. All of his comments are reasonable for me, and I agree with his comments.

Author Response

(The authors gave the same response as above.)
